# The *Trypanosoma brucei* MISP family of invariant proteins is co-expressed with BARP as triple helical bundle structures on the surface of salivary gland forms, but is dispensable for parasite development within the tsetse vector

Aitor Casas-Sanchez[1,2], Raghavendran Ramaswamy[3°], Samïrah Perally[2°], Lee R. Haines[1¤a], Clair Rose[1], Marcela Aguilera-Flores[4], Susana Portillo[4¤b], Margot Verbeelen[5], Shahid Hussain[5], Laura Smithson[6¤c], Cristina Yunta[1], Michael J. Lehane[1], Sue Vaughan[6], Jan van den Abbeele[5], Igor C. Almeida[4], Martin J. Boulanger[3*], Álvaro Acosta-Serrano[1,2¤a*]

**1** Department of Vector Biology, Liverpool School of Tropical Medicine, Liverpool, United Kingdom, **2** Department of Tropical Disease Biology, Liverpool School of Tropical Medicine, Liverpool, United Kingdom, **3** Department of Biochemistry and Microbiology, University of Victoria, Victoria, Canada, **4** Border Biomedical Research Center, Department of Biological Sciences, University of Texas at El Paso, El Paso, Texas, United States of America, **5** Institute of Tropical Medicine, Antwerp, Belgium, **6** Biological and Medical Sciences, Oxford Brookes University, Oxford, United Kingdom

° These authors contributed equally to this work.
¤a Current address: Department of Biological Sciences, Notre Dame University, South Bend, Indiana, United States of America
¤b Current address: Center for Vaccine Development and Global Health, University of Maryland School of Medicine, Baltimore, Maryland, United States of America
¤c Current address: Department of Molecular Microbiology and Immunology, Brown University, Providence, Rhode Island, United States of America
* mboulang@uvic.ca (MJB); aacosta3@nd.edu (AAS)

## Abstract

*Trypanosoma brucei* spp. develop into mammalian-infectious metacyclic trypomastigotes inside tsetse salivary glands. Besides acquiring a variant surface glycoprotein (VSG) coat, little is known about the metacyclic expression of invariant surface antigens. Proteomic analyses of saliva from *T. brucei*-infected tsetse flies identified, in addition to VSG and Brucei Alanine-Rich Protein (BARP) peptides, a family of glycosylphosphatidylinositol (GPI)-anchored surface proteins herein named as Metacyclic Invariant Surface Proteins (MISP) because of its predominant expression on the surface of metacyclic trypomastigotes. The MISP family is encoded by five paralog genes with >80% protein identity, which are exclusively expressed by salivary gland stages of the parasite and peak in metacyclic stage, as shown by confocal microscopy and immuno-high resolution scanning electron microscopy. Crystallographic analysis of a MISP isoform (MISP360) and a high confidence model of BARP revealed a triple helical bundle architecture commonly found in other trypanosome surface proteins. Molecular modelling combined with live fluorescent microscopy suggests that MISP N-termini are potentially extended above the metacyclic VSG coat, and thus could

**Data Availability Statement:** All relevant data are within the manuscript and its Supporting information files.

**Funding:** This work was partially supported by a project grant (093691/Z/10/Z; awarded to AAS) and a Multi-User Equipment Grant (104936/Z/14/Z) from the Wellcome Trust (http://www.wellcome.ac. uk), GlycoPar-EU FP7 Marie Curie Initial Training Network (GA. 608295; http://www.ec.europa.eu; awarded to ACS and AAS), the Medical Research Council Concept in Confidence scheme MC_PC_17167; https://www.ukri.org/councils/ mrc/; awarded to AAS), the Natural Sciences and Engineering Research Council of Canada (https:// www.nserc-crsng.gc.ca; awarded to MJB), and a grant from The National Institute on Minority Health and Health Disparities (NIMHD) (https:// www.nimhd.nih.gov/) for supporting The Biomolecule Analysis and Omics Unit (BAOU) at UTEP/BBRC. The funders had no role in study design, data collection and analysis, decision to publish, or preparation of the manuscript.

**Competing interests:** The authors have declared that no competing interests exist.

be tested as a transmission-blocking vaccine target. However, vaccination with recombinant MISP360 isoform did not protect mice against a *T. brucei* infectious tsetse bite. Lastly, both CRISPR-Cas9-driven knock out and RNAi knock down of all MISP paralogues suggest they are not essential for parasite development in the tsetse vector. We suggest MISP may be relevant during trypanosome transmission or establishment in the vertebrate's skin.

## Author summary

The *Trypanosoma brucei* group of parasites are exclusively transmitted to the vertebrate host by the tsetse vector alongside insect saliva. To better understand trypanosome transmission, we investigated the protein composition of *T. brucei*-infected tsetse saliva using a mass spectrometry proteomics approach. We found that, in addition to proteins from tsetse saliva and *Sodalis glossinidius* (a bacterial tsetse symbiont), trypanosome-infected saliva contains several parasite surface glycoproteins, including a partially characterized family of invariant proteins herein named Metacyclic Invariant Surface Protein (MISP). We show that MISP is primarily expressed, together with metacyclic Variant Surface Glycoprotein (mVSG) and Brucei Acidic Repetitive Protein (BARP), on the surface of the infectious metacyclic stage of *T. brucei*. Its triple helix bundle architecture appears tethered to the outer membrane by an extended glycosylphosphatidylinositol-anchored C-terminal tail that putatively projects MISP above the VSG coat. Our findings provide new insights into the surface architecture of the *T. brucei* metacyclic stage and describe the challenges associated with developing transmission-blocking vaccines against tsetse-transmitted trypanosomes.

## Introduction

African trypanosomes are the causative agents of human African trypanosomiasis (HAT) or sleeping sickness and animal African trypanosomiasis (AAT or *Nagana* disease) in most sub-Saharan countries. Multiple trypanosome species, including the group of *Trypanosoma brucei* parasites (i.e. *T. b. gambiense*, *T. b. rhodesiense*, and *T. b. brucei*), are transmitted by several tsetse species (*Glossina* spp.). Despite decreasing HAT cases since 2010, due largely to a program of active screening and treatment of the human population together with an efficient vector control program, AAT continues to cause significant economic losses in the agricultural sector with >1 million cattle dying each year [1]. Since these parasites undergo antigenic variation to effectively evade the host's immune system, preventive vaccines have been difficult to develop, and only one has recently shown protection against *T. vivax* [2].

During its life cycle, *T. brucei* undergoes many developmental changes to adapt to the different environments within the mammalian host and the tsetse vector [3]. All *T. brucei* life stages have in common the expression of major surface glycosylphosphatidylinositol (GPI)-anchored glycoproteins known to be important for parasite survival. In bloodstream form (BSF) stages, the variant surface glycoprotein (VSG) coat is responsible for antigenic variation [4–6], clearance of host immunoglobulins bound to VSGs [7] and preventing antibodies from binding to conserved VSG epitopes and invariant surface proteins (ISG), such as transporters and receptors [8–11]. When 'stumpy' BSF trypanosomes are ingested by tsetse, they quickly (24 to 48 hours) differentiate into the procyclic trypomastigote stage within the fly's midgut [3,12]. During differentiation, VSGs are no longer expressed and the parasites lose the VSG coat via the concerted action of major surface metalloproteases and the GPI-phospholipase C

(GPI-PLC) [13,14]—a process that appears to modulate the tsetse's immunity in favor of parasite infection [15]. Concomitant with VSG disappearance, the parasites express a new coat comprised of GPEET- and EP-procyclin isoforms once the infection establishes in the midgut [16–18] and proventriculus (PV) [19]. Within the tsetse midgut, procyclic trypomastigotes first colonize the ectoperitrophic space of the anterior midgut, from which they further migrate to the tsetse cardia or PV [20]; alternatively, both the anterior midgut and PV can be simultaneously colonized as trypanosomes reach the ectoperitrophic space after crossing an immature peritrophic matrix at the PV [19]. After 10–15 days post-infection, depending on parasite strain, trypanosomes differentiate into epimastigote forms (EMFs) within the PV, which then migrate to the salivary glands (SGs), attach to the epithelial cell microvilli and switch on the expression of Brucei Alanine-Rich Proteins (BARP) [21]. The attached EMFs further differentiate into pre-metacyclic trypomastigotes [22,23], during which time BARP expression is lost, a new coat of metacyclic VSG (mVSG) develops [24,25], and cells detach from the SG epithelium. Finally, mature metacyclic trypomastigote forms (MCFs) [23,26] are inoculated along with saliva into the skin of a vertebrate host in a subsequent feed. Unlike in BSFs, the surface composition of *T. brucei* MCFs is less well characterized primarily due to the lack of an efficient *in vitro* culture system of this life stage, the low percentage of flies with salivary gland infections, and the low yields of MCFs from infected salivary glands.

In contrast to BSFs that possess a large repertoire of *vsg* genes [27,28], MCFs express a single mVSG protein (also called metacyclic variant antigen type; mVAT) from a much-reduced gene repertoire. Based on antibody recognition, it is estimated that a population of tsetse-derived MCFs express different mVSGs (as many as 27) to ensure infection of hosts that may have been pre-exposed to any of these antigenic types [29,30]. Given that MCFs develop in the tsetse SGs and their persistence in the host after transmission is relatively short [26], they do not seem pressured to undergo antigenic variation. Despite the differences in the expression mechanisms between VSGs and mVSGs, it is assumed that their function and structural organization is equivalent. BSF VSGs form a protective outer layer that covers the parasite surface (~5 million homodimers/cell) [31] to mask the invariant surface proteins from the host immune system. Many of these invariant proteins are predicted to be shorter in height than the VSG homodimers and to be attached to the plasma membrane via transmembrane domains [8,11]. The exceptions to this are the haptoglobin-hemoglobin (HpHbR) and transferrin receptors, which are GPI-anchored and exclusively localize in the flagellar pocket [9,10].

During a blood meal, the infected tsetse inoculates MCFs into the host along with saliva [26,32] to inhibit the host hemostatic response to the cutaneous trauma [33,34]. However, during parasite development in the tsetse SGs, there is a drastic (~80%) transcriptional down regulation of genes encoding for salivary proteins, which induces a feeding phenotype that may amplify disease transmission [32,35,36]. Although the proteome of tsetse saliva from *T. brucei*-infected flies has been determined [34], soluble parasite factors have not been characterized in detail and thus, the molecular mechanism by which these parasites manipulate the vector to promote transmission is largely unknown. Here, we identified trypanosome proteins from infected tsetse saliva using a high-resolution mass spectrometry-based proteomics approach. We found that *T. brucei*-infected saliva is particularly enriched with peptides from trypanosome surface proteins such as BARP, mVSG and a partially characterized family of glycoproteins herein named Metacyclic Invariant Surface Proteins (MISP). We show evidence that MISP, previously named as salivary gland epimastigote 1 (SGE1) [37, 38], are a small family of invariant glycoproteins primarily expressed on the surface of the metacyclic stage of *T. brucei*. Structural data reveal a triple helix bundle architecture for MISP that appears to be tethered to the outer membrane by an extended, unstructured C-terminal tail, potentially projecting MISP above the mVSG coat and thereby allowing antibody recognition. Our findings provide

a new insight into the surface architecture of MCFs and are discussed with respect to the molecular crosstalk between parasite and vector, and the challenges associated with developing transmission-blocking vaccines against tsetse-transmitted trypanosomes.

## Results

### Proteomic analysis of *T. brucei*-infected tsetse saliva reveals an enrichment of parasite GPI-anchored surface proteins

We initially investigated the protein composition of *Glossina morsitans morsitans* saliva with a *T. b. brucei* infection in the salivary glands to identify novel soluble factors important for parasite transmission. We performed high-resolution nano-liquid chromatography-tandem mass spectrometry (nLC-MS/MS) on pools of saliva from flies at 30 days post-infection (dpi) in three biological replicates (S1 Fig). After filtering out the *Glossina* protein hits (S1 Table), we detected peptides from the bacterial symbiont *Sodalis glossinidius* [39] in both uninfected and trypanosome-infected saliva (S2 Table). The presence of these symbiont peptides was validated by immunoblotting using an anti-*Sodalis* Hsp60 monoclonal antibody (S2 Fig). Following manual curation of the MS/MS spectra, we also identified 45 unique peptides derived from a total of 27 *T. b. brucei* proteins in infected saliva datasets only (>95% confidence protein identification) (S3 Table). Of the identified proteins, 17 (62.9%) were predicted to be intracellular (S4 Table) mainly linked to functions such as protein folding (14.8%), cytoskeleton (11.1%), and metabolic processes (7.4%). The remaining 10 proteins (37.1%) were predicted to be GPI-anchored surface proteins from three major families (Fig 1A). We found four unique peptides shared by eight different BARP isoforms and identified five different mVSGs from seven unique peptides (Fig 1A, S1 File and S3 Table). Notably, we also identified a single peptide (100% confidence, supported by two spectra, S1 File and S3 Table) belonging to SGE1 [37], a small family of GPI-anchored proteins we characterized and renamed metacyclic invariant surface proteins (MISP) (Fig 1A) because of its predominant life-cycle expression (see below). The conserved peptide 'SVAEDNSAASTAR' was fully conserved across all MISP isoforms (Fig 1B–1D and S3 Table).

### The Metacyclic Invariant Surface Proteins (MISP) family

MISP proteins are part of the large Fam50 family of trypanosome surface proteins [40]. *T. b. brucei* contains five *misp* paralog genes (*Tb927.7.360*, *Tb927.7.380*, *Tb927.7.400*, *Tb927.7.420*, *Tb927.7.440* in reference strain TREU927), which encode for proteins that are unique in sequence and have no predicted function. All five *T. b. brucei* paralogs are localized to chromosome 7 (Fig 1C) and are separated by unrelated genes encoding hypothetical proteins (*Tb927.7.370*, *Tb927.7.390*, *Tb927.7.410* and *Tb927.7.430*). Notably, the same *misp* homologs are found in *T. b. gambiense* (*Tbg972.7.270* and *Tbg972.7.290*). Furthermore, slightly divergent versions of *misp* genes are found in the animal trypanosome species *T. evansi* (*TevSTIB805.7.300*, *TevSTIB805.7.320* and *TevSTIB805.7.380*) and *T. congolense* (*TcIL3000.0.02370*), and no homologs were found in the *T. vivax* genome (S3 Fig). Using a phylogenetic analysis (PhyML 3.1) [41], we determined that MISP are comprised of two sub-families (MISP-A and MISP-B) (Fig 1B). MISP-A members are mainly characterized by containing one to three 26 residue repetitive motives at the C-terminus (see below) and MISP-B isoforms, in contrast, present no C-terminal repeats. The *T. congolense* homolog (TcIL3000.0.02370) shares only 34.2% identity with the others and was not included into either sub-family (S3B Fig). A comparison of all *T. b. brucei* MISP paralogs revealed an overall amino acid sequence identity of ~79% (Fig 1D), with the N-terminal domain (Met$^1$–Glu$^{237}$) showing greater conservation (88.7% identity) compared to the divergent

**A**

| Protein ID | Annotation | Peptides | Spectra |
|---|---|---|---|
| Tb927.9.15520 [I] | Brucei Alanine-Rich Protein (BARP) | 1 | 3 |
| Tb927.9.15530 | Brucei Alanine-Rich Protein (BARP) | 4 | 7 |
| Tb927.9.15570 | Brucei Alanine-Rich Protein (BARP) | 1 | 2 |
| Tb927.9.15630 | Brucei Alanine-Rich Protein (BARP) | 1 | 1 |
| VSM2_TRYBB | Variant Surface Glycoprotein (VSG) 221 | 3 | 4 |
| O76421_TRYBR | Metacyclic Variant Antigen Type (mVAT) 4 | 1 | 1 |
| A0A1J0RB71_9TRYP | Variant Surface Glycoprotein (VSG) 1125.4959 | 1 | 1 |
| A0A1J0R978_9TRYP | Variant Surface Glycoprotein (VSG) 1125.3088 | 1 | 1 |
| A0A1J0RAQ3_9TRYP [II] | Variant Surface Glycoprotein (VSG) 1125.408 | 1 | 1 |
| Tb927.7.360 [III] | Metacyclic Invariant Surface Protein (MISP) | 1 | 2 |

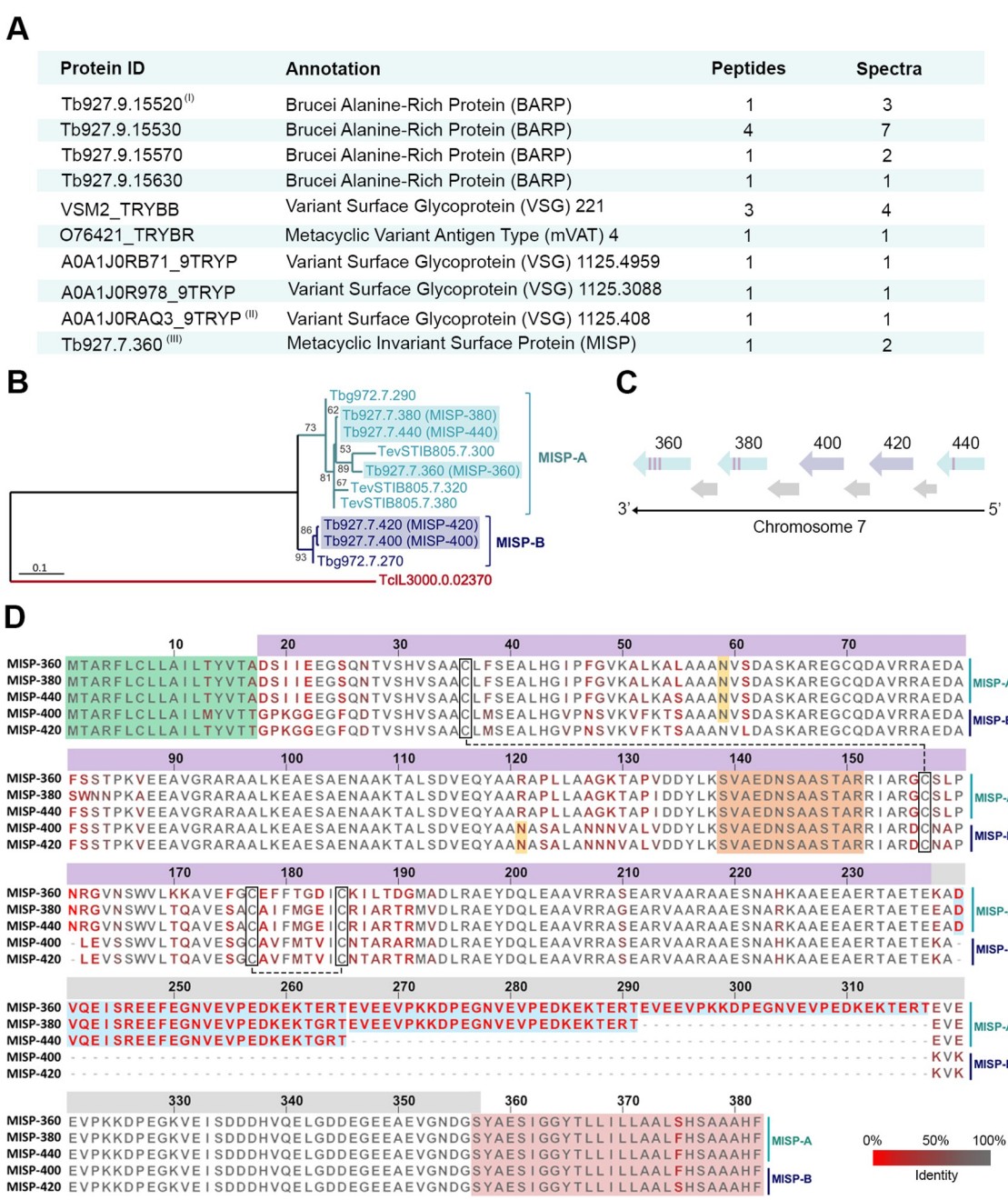

**Fig 1. Identification of sequence analyses of MISP. (A)** Summary table of trypanosome GPI-anchored proteins identified in *T. brucei*-infected tsetse saliva by nLC-MS/MS. All identified proteins and peptides have >95% confidence and were manually curated. Protein accession code (Protein ID), protein description (Annotation), number of identified unique peptides (Peptides), and number of spectra (Spectra) are indicated; see peptides and spectra in S1 File. Accession codes either from TriTrypDB (BARP and MISP) or UniProt (VSG). (I) representative ID for Tb927.9.15520, Tb927.9.15530, Tb927.9.15550, Tb927.9.15590, Tb927.9.15600, Tb927.9.15620 and Tb927.9.15660; (II) representative ID for VSGs 646, 769, 3613, 1125.408, 1125.474, 1125.1142, 1125.4207 and 1125.4707; (III) representative ID for Tb927.7.360, Tb927.7.380, Tb927.7.400, Tb927.7.420, Tb927.7.440. **(B)** Phylogenetic tree generated from a multiple sequence alignment of all MISP predicted proteins. Note the demarcation of two sub-families MISP-A (light blue) and MISP-B (dark blue) and the least-conserved *T. congolense* isoform (red). The *T. b. brucei* MISP are boxed. Tree created using PhyML 3.1 and the WAG substitution model of maximum likelihood; bootstrap of 500; numbers show branch support values (%); scale bar: 0.1 substitutions/site. **(C)** Schematic representation of the *T. b. brucei* chromosome 7 region containing the *misp* gene array. Genes represented by arrows; *misp-A* (light blue), *misp-B* (light purple), unrelated genes (light grey); C-terminal repeats represented by purple rectangles on the arrows. **(D)** Multiple protein alignment of *T. b. brucei* MISP isoforms. Residues colored from grey (full conservation) to bright red (highest divergence). Protein domains highlighted: N- signal peptide (green),

predicted *N*-glycosylation sites (yellow asparagine), disulphide bonds identified in crystal structure (cysteines in black boxes), C-terminus repetitive motifs of 26 residues (blue) and GPI-anchor signal peptide (red). The conserved peptide 'SVAEDNSAASTAR' identified by nLC-MS/MS is highlighted in orange. The top bar numbers indicate residue position starting from Met[1]. Top bar colors indicate the main protein domains N-terminus (purple) and C-terminus (grey).

C-terminal (Lys$^{238}$-Ser$^{357}$) (49.0% identity). Importantly, sequence divergence is not random but primarily localized to the charged 26 residue repetitive motifs (EVEEVPKKDPEGNVEVP EDKEKTERT or DVQEISREEFEGNVEVPEDKEKTERT). We identified predicted signal peptides for all MISP isoforms (Met$^1$–Ala/Thr$^{17}$). While no transmembrane domains were identified, all MISP isoforms contain a predicted GPI omega site at the C-terminus (Fig 1D and S5 Table). Furthermore, all MISP isoforms contain at least one potential *N*-glycosylation site (Asn$^{59}$), while Tb927.7.400 and Tb927.7.420 contain an additional site on Asn$^{121}$ (S5 Table). Overall, sequence-based predictions suggest that MISP is a family of surface GPI-anchored glycoproteins presenting little sequence divergence and differential C-terminal lengths.

## MISP are expressed on the surface of *T. b. brucei* salivary gland stages

To understand the expression pattern of individual *misp* paralogs throughout the *T. b. brucei*'s life cycle, we used semi-quantitative RT-PCR to detect transcript levels of individual *misp* isoforms and *barp*, which are markers for salivary gland epimastigotes [21] (Fig 2A). Total RNA was purified from all isolated *T. b. brucei* life stages obtained from both *in vitro* cultures (i.e., bloodstream forms and procyclic cultured forms) and tsetse-derived parasites at 30 dpi. (i.e., midgut procyclics, proventricular parasites (mixture of 95% mesocyclics and 5% epimastigotes), whole infected SGs, and metacyclics isolated from infected saliva (mixture of 95% metacyclics and 5% epimastigotes). Thirty day old uninfected flies were used as an age-matched control. Due to the high sequence identity among *misp* paralogs, assessing the expression of individual *misp* transcripts by real-time RT-PCR was not feasible and hence, we used semi-quantitative RT-PCR to target the different C-terminal *misp* repetitive regions, which are unique amongst four of the paralogs with differences of 78 bp (S4 Fig). While all control samples from uninfected flies were negative, whole infected SGs and purified metacyclics from saliva showed the highest levels of *misp* transcripts compared to midgut procyclics (11.8-fold) and BSFs (25.1-fold) (Fig 2A). Expression of *misp* transcripts was also found to be high in trypanosomes colonizing the PV, suggesting it is in those stages where gene up regulation starts. Although there seems to be an overall preferential selection for paralogs with the shortest C-termini (*misp400/420*), the longer paralogs *misp380* and *misp360* were up regulated and significantly higher in SG stages. In parallel, the transcriptional profile of *misp* was validated by measuring that from a transgenic cell line overexpressing *misp360* under a tetracycline-inducible promoter (S4 Fig). Interestingly, all *barp* transcripts (originated from 14 paralogs sharing >60% identity, S5 Fig) followed a similar expression pattern, although PV-associated trypanosomes were the highest expressors, suggesting that the expression of *barp* may precede that of *misp* genes during development.

Having identified SG stages as the major expressors of *misp* and *barp* transcripts, we next determined which life stages translated these transcripts into proteins by immunostaining with polyclonal antibodies that recognize epitopes conserved across all MISP or BARP isoforms. Antibody binding was only to SG parasites, both within the glands (S6 Fig) and *ex vivo* (Fig 2B and 2C), and no recognition was detected on BSFs, PCFs, midgut procyclics or PV parasites by either antibody (S7 Fig). Surprisingly, despite PV-associated trypanosomes showing high levels of *misp* and *barp* transcripts, the corresponding gene products are not produced until they reach the next developmental stage in the salivary glands. Instead, we confirmed that

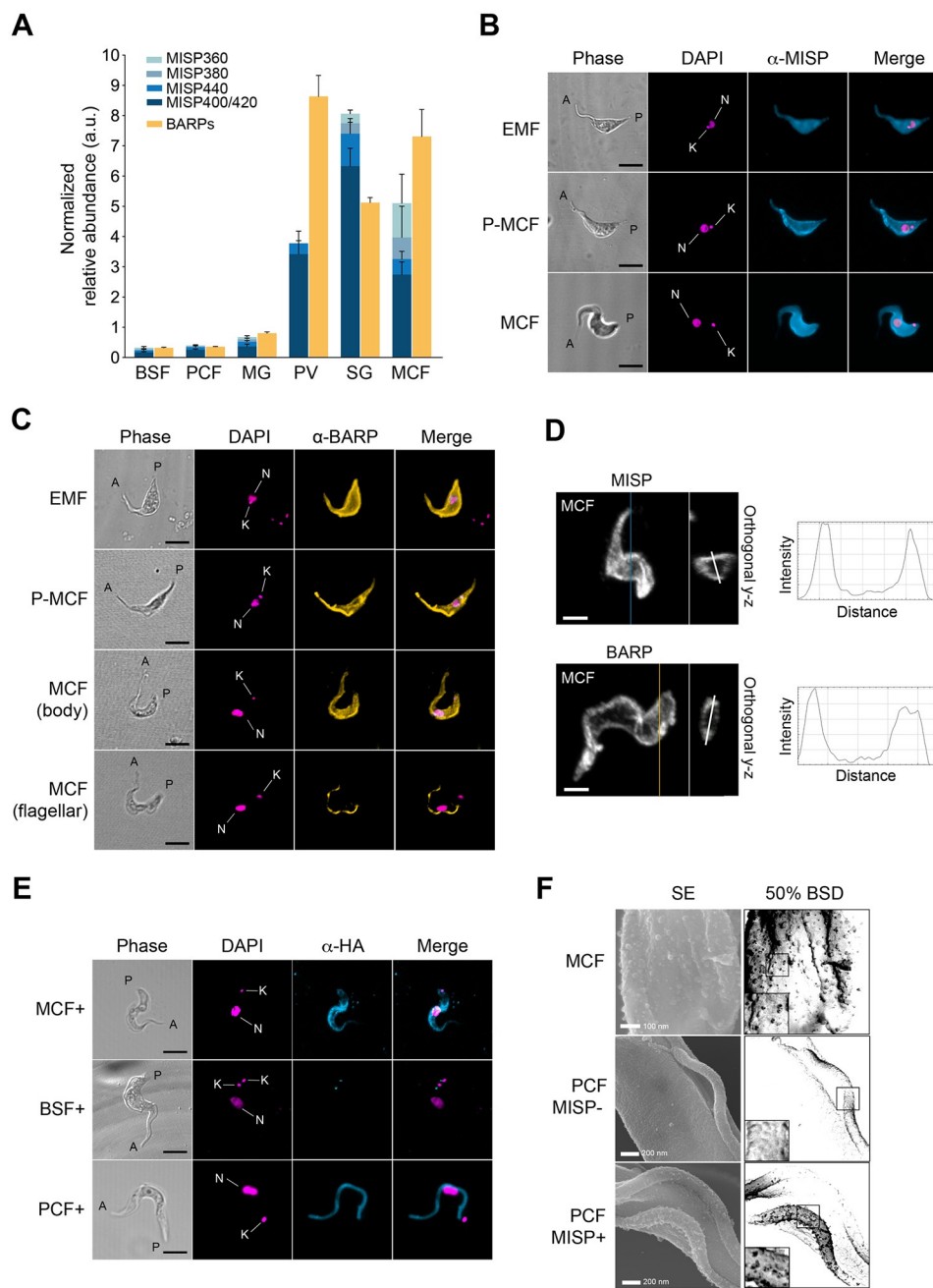

**Fig 2. Expression and localization of MISP and BARP.** (**A**) Relative mRNA expression levels (arbitrary units, normalized to the expression of *tert*) of *T. b. brucei misp* paralogs (blue) and *barp* (yellow), in cultured bloodstream forms (BSF), procyclic cultured forms (PCF), midgut procyclics (MG), proventricular forms (PV), salivary gland forms (SG) and purified metacyclic forms (MCF). Bars represent mean values; error bars represent ±S.D from two technical replicates. (**B**) Representative images of non-permeabilized fixed parasites extracted from infected tsetse salivary glands, immunostained with either anti-MISP (**B**) or anti-BARP (**C**) polyclonal antibodies; epimastigote (EMF), pre-metacyclic (P-MCF) and metacyclic forms (MCF); shown as phase, anti-MISP (cyan) or anti-BARP (yellow), and DAPI (magenta); kinetoplasts (K) and nuclei (N) are annotated in the DAPI channel; anterior (A) and posterior (P) ends of the parasite are annotated in the Phase channel. (**D**) Maximum intensity projection of representative metacyclic cells immunostained with anti-MISP (top) or anti-BARP (bottom) polyclonal antibodies and 3D-reconstructed from z-stacks; y-z orthogonal views show surface localization of the signal and graphs on the right plot fluorescence intensity over the white line crossing the cell section. (**E**) Localization of anti-HA monoclonal antibodies on metacyclic (MCF), bloodstream form (BSF) and procyclic cultured form (PCF) transgenic *T. b. brucei* cells overexpressing ectopic HA-tagged MISP360 upon doxycycline induction; shown as phase, anti-HA (cyan), DAPI

(magenta) and merge; kinetoplasts (K) and nuclei (N) are annotated in the DAPI channel; anterior (A) and posterior (P) ends of the parasite are annotated in the Phase channel. (**F**) Representative HRSEM images of wild type metacyclic (MCF), procyclic MISP360 overexpressor uninduced (PCF MISP -) and induced (PCF MISP+) cell surfaces immunostained with anti-MISP polyclonal antibodies; secondary electron detection (SE), 50% inverted backscatter electron detection (50% BSD); scale bars annotated in SE.

both mesocyclic and epimastigote forms in the proventriculus expressed EP-procyclin as the major surface GPI-anchored protein (S8 Fig). Furthermore, both polyclonal anti-MISP and anti-BARP antibodies recognized all parasite cells and stages in infected SGs, including attached epimastigotes, pre-metacyclics and mature metacyclic forms (Figs 2B, 2D and S6). The relative mean fluorescence intensity of MISP was higher in metacyclics compared to epimastigotes (2.3-fold); in contrast, BARP intensity was found to be higher in epimastigotes than in metacyclics (3.3-fold) (S9 Fig). Moreover, MISP appeared to localize evenly across the cell surface of non-permeabilized fixed epimastigote and metacyclic cells (Fig 2B). In contrast, BARP appears homogeneously expressed on the surface of all epimastigotes, yet on metacyclics, BARP is detected on the surface of few (~10.4% of the cells; n = 481), while the rest show exclusively flagellar expression (Figs 2C and S10). To confirm the identity of SG-specific trypanosomes, we measured the length of parasites and the relative position of kDNA and nucleus (S11 Fig), and performed immunostainings with anti-CRD polyclonal antibodies, which recognize the GPI cross-reacting determinant (CRD) epitope [23,42]. While epimastigotes and pre-metacyclics were not detected by the antibody, mature metacyclics showed surface staining as indirect detection of mVSGs that are partially cleaved by GPI-PLC (S12 Fig). To confirm the localization of MISP and BARP in metacyclics, immuno-stained cells were 3D-rendered using confocal laser scanning microscopy and orthogonal views from z-stacks confirmed the fluorescence only localized to the edges of the cell body (Fig 2D).

As an additional approach to investigate MISP localization, we engineered a *T. b. brucei* transgenic cell line to ectopically express HA/eGFP-tagged MISP360 under a tetracycline-inducible promoter (Figs 2E and S13). Upon induction, ectopic MISP localized on the metacyclic cell body surface in a pattern resembling that of wild type cells (S13C Fig). However, in BSF, ectopic MISP was only detected in the flagellar pocket and, upon differentiation to PCF, it was then found redistributed to the flagellum and co-localised with the paraflagellar rod (PFR) (S13C Fig). We also used high-resolution scanning electron microscopy (HRSEM) to precisely locate anti-MISP antibodies on the surface of stained cells (Fig 2F). While uninduced MISP360 overexpressing PCF cells did not show antibody binding, induction of the transgene led to detection of the ectopic MISP mainly along the flagellum as in Fig 2E. In contrast, native MISP was equally detected in wild type metacyclic cells on both flagellum and cell body, thus validating the observations from confocal microscopy (Fig 2B and 2D). To confirm exposure of MISP epitopes in native conditions and free of potential fixative-derived artifacts, we performed immunostaining on live wild type metacyclic cells, which still displayed positive binding of anti-MISP antibodies on the cell surface (S14 Fig). Lastly, metacyclics extracted from flies infected with *T. brucei* TSW196 strain are also recognized by the anti-MISP antibody (S15 Fig), confirming that staining is not specific for AnTat 1.1 metacyclic trypanosomes.

## The N-terminal ectodomains of MISP and BARP adopt a triple helix bundle structure

To complement the expression and localization analyses, we next determined the crystal structure of the conserved N-terminal ectodomain (Gly$^{24}$-Ala$^{234}$) of MISP360 (Tb427.07.360; *T. b. brucei* MISP360 homolog in reference strain Lister 427) to 1.82 Å resolution (Fig 3). Crystals

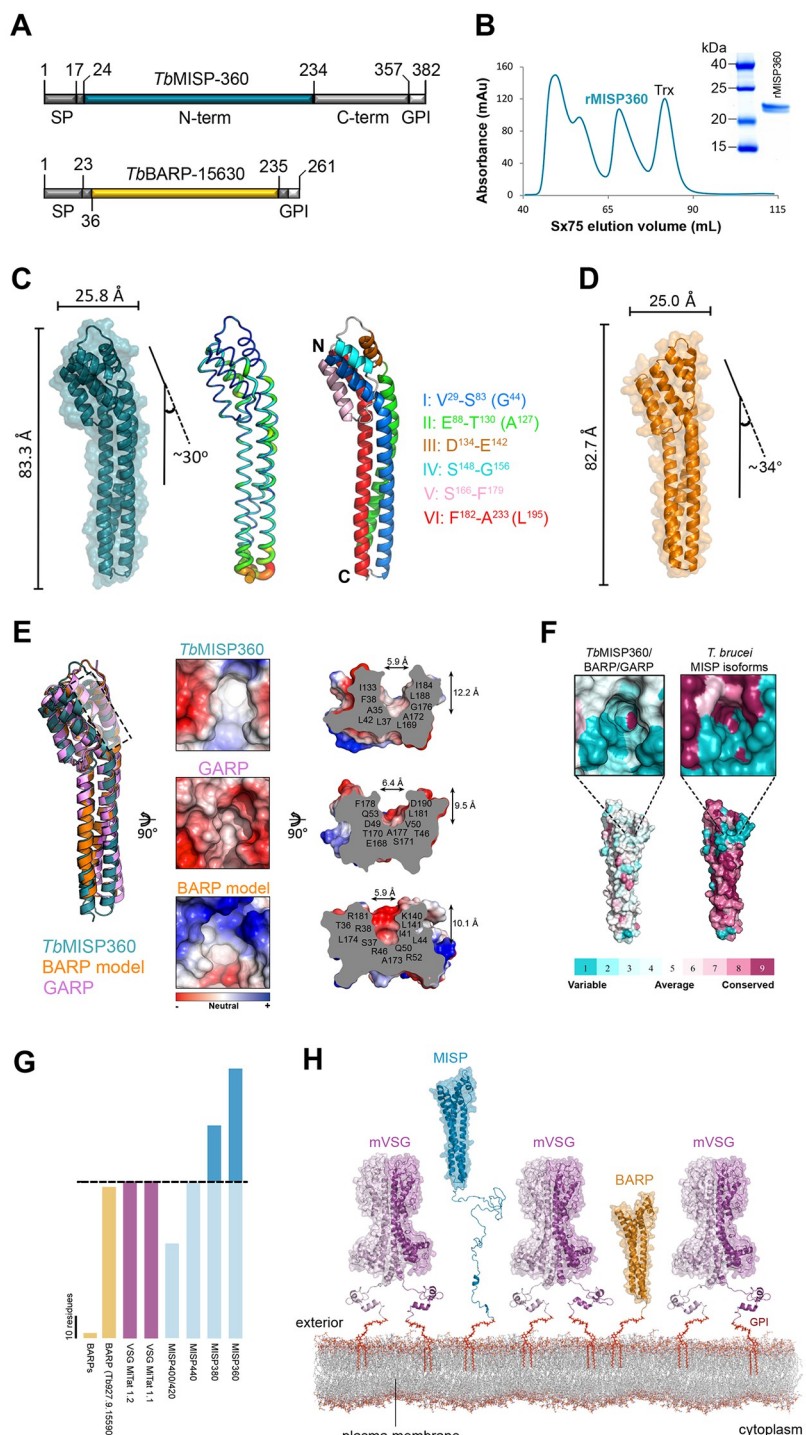

**Fig 3. Crystal structures of MISP and BARP N-terminal ectodomains.** (**A**) Schematic representation of MISP360 and BARP protein sequences, highlighting in color the region recombinantly expressed in *E. coli* and crystallized; numbers indicate residue positions, signal peptide (SP), C-terminal domain (C-term), GPI-anchor peptide (GPI). (**B**) FPLC SEC elution chromatogram of recombinant MISP360; SDS-PAGE of monomeric recombinant MISP360 (inset). (**C**) MISP360 N-terminus crystal structure with molecular surface representation. The structure was found to be 83.3 Å tall, 25.5 Å in width with a 30° helix bend at the top (left); *B*-factor putty model with ordered regions displayed in blue and thin tubes and flexible regions in red and thicker tubes (middle); secondary structure depiction highlighting the organization of the helices (right); bend-forming residues are in parentheses; coiled structures are represented in white. (**D**) BARP high confidence homology model generated with Modeller 9v18 [43] shown with a semi-transparent surface

representation. The BARP model reveals an overall architecture that is 82.7 Å tall, 25.0 Å in width with a 34° helix bend consistent with the molecular envelope calculated from the 3.2 Å X-ray crystallography diffraction data set. (**E**) Structural superimposition of the MISP360 N-terminus crystal structure (cyan) with GARP (purple, PDB: 2Y44) and the high confidence model of BARP (orange) (left). A black dashed line highlights the area where the apical surface pocket is found. Top view of an electrostatic surface representation of MISP360, BARP and GARP where the surface pocket is found (middle), and its orthogonal view indicating the residues forming the pocket and size (right). (**F**) Conservation of the pocket-forming (top) and overall (bottom) residues across MISP360, BARP and GARP depicted on the MISP360 structure (left). Residue conservation between the five *T. brucei* MISP isoforms on the MISP360 structure (right); from most variable (dark blue) to most conserved (dark purple). (**G**) Schematic comparison of the C-terminal domain lengths of BARP, VSG and MISP; scale bar = 10 residues; dashed line sets length baseline (VSG). (**H**) Structural model of the proposed *T. brucei* metacyclic surface glycocalyx displaying the GPI-anchored metacyclic VSG homodimers (mVSG; light and deep purple), MISP-A (cyan), MISP-B (dark blue) and remains of BARP (yellow). The mVSG structure is represented with a model of mVAT4 based on the crystal structure of VSG 221 N- (PDB: 1VSG) and C-terminus (PDB: 1XU6). MISP is represented with the crystal structure of MISP360 N-terminus (PDB: 5VTL) and a model of its C-terminus; BARP is represented with our confidence model.

of purified, monomeric MISP360 (Figs 3A–3B and S16A–S16B) were obtained using the hanging drop method and grew in space group P2$_1$2$_1$2$_1$ with a single molecule in the asymmetric unit. The structure of MISP360 was determined by molecular replacement using a truncated form of *T. congolense* GARP (PDB entry: 2Y44) as the search model. The identification of GARP as a suitable model was based on secondary structure predictions as the amino acid sequence identity of the N-terminal ectodomain is only 15%. The overall structure of MISP360 was well defined with only two residues from the N-terminus remaining unmodelled (Gly$^{24}$ and Ser$^{25}$). The core of the N-terminal ectodomain adopts an elongated structure measuring approximately 83 Å in height and spanning approximately 26 Å in width (Fig 3C, left). The ectodomain is well ordered with low *B*-factors throughout the protein (Fig 3C, middle). MISP360 adopts an overall triple helical bundle structure composed of a core of extended twisted helices capped by a smaller helical bundle at the N-terminal end. The helical bundle that dominates the structure consists principally of three helices (Fig 3C, right): helix I is comprised of residues Val$^{29}$-Ser$^{83}$, helix II of Glu$^{88}$-Thr$^{130}$ and helix VI of Phe$^{182}$-Ala$^{233}$. The three helices adopt a bend of approximately 30 degrees at Gly$^{44}$ (helix I), Ala$^{127}$ (helix II) and Leu$^{195}$ (helix VI) and collectively form the helical bundle cap. In addition to the terminal portions of the three major helices, the helical bundle includes three shorter helices: helix III (Asp$^{134}$-Glu$^{142}$), which is connected by a 5-residue loop to helix IV (Ser$^{148}$-Gly$^{156}$), and helix V (Ser$^{166}$-Phe$^{179}$) connected to helix IV by a 9-residue loop. Helices IV and V lie at either side of helix I and form the broadest face of the helical bundle cap. Moreover, the head structure of MISP360 is anchored by two disulphide bonds; one between Cys$^{36}$ and Cys$^{157}$, and the other between Cys$^{177}$ and Cys$^{185}$ (Fig 1D).

Given that BARP is co-expressed with MISP in both epimastigote and metacyclic forms (Fig 2C), we also sought to determine its crystal structure after expression in *E. coli* and purification (S16C and S16D Fig). Diffraction data was collected to 3.2 Å resolution and structure solution revealed a molecular envelope consistent with MISP. However, the data quality did not support a detailed atomic level refinement and thus we opted to generate a high-confidence BARP (Tb927.9.15630) homology model using Modeller 9v18 [43]. Consistent with the X-ray data, the BARP model revealed an extended helical architecture with comparable dimensions and a largely conserved helical bundle relative to MISP360 (Fig 3D).

The lack of significant sequence identity between MISP360 and any protein with known function led us to perform a DALI search [44] to identify structural homologs. *T. congolense*'s GARP, of unknown function, was identified as the top hit with a Z-score of 21.4. A least squares superposition between the two structures resulted in a root-mean-square deviation of 1.7 Å over 183 C$_\alpha$ atoms (Fig 3E, left). The DALI search also revealed structural homology to

the previously characterized haptoglobin-hemoglobin receptor (HpHbR) from *T. brucei* (PDB: 4X0J) and *T. congolense* (PDB: 4E40) [45, 46], and to a *T. brucei* VSG monomer (PDB: 1VSG) with Z-scores of 18.1, 17.7 and 6.5, respectively. All these proteins exhibit a complementary core of twisted three helical bundles (S17 Fig). Structural comparison, however, indicates a closer architectural similarity with HpHbR compared to VSG, due in part to the breakdown of the third helix into loops and extensions enabling substantial conformational diversity in VSG [47]. Moreover, MISP360 is monomeric in contrast to dimeric VSGs [47]. Despite the general structural similarity with GARP, BARP, HpHbR and the VSG monomer, the overall low sequence identity and the lack of key conserved residues suggest a different biological role for MISP.

Surface analysis of the MISP360 and GARP [48] crystal structures, and the BARP high confidence homology model revealed a similar distribution of acidic and basic patches along the entire length of the structure with no clear localized charge densities that would indicate a molecular recognition site (Fig 3E). However, a surface pocket of similar dimensions was identified at the membrane distal end near the region where the core helices bend in each protein (Fig 3E), but intriguingly, the residues forming the pockets are quite divergent. While the pocket in MISP360 is mainly formed by hydrophobic residues, BARP's contain both hydrophobic and basic residues, and GARP's are formed by acidic, polar and hydrophobic amino acids. This variability in pocket residue composition prompted us to map the residues conserved between MISP360, BARP and GARP onto the MISP360 crystal structure using ConSurf [49, 50] (Fig 3F, left), which showed no clear conservation among the pocket-forming residues. Moreover, mapping conservation between the five *T. brucei* MISP isoforms on MISP360, revealed that most of the conserved residues are found forming the core of the helices and pocket. However, a striking 51% of the non-conserved residues (representing 22.3% of the full protein sequence) are in close proximity to the pocket (Fig 3F, right), suggesting these residues may mediate differential affinity or recognition of putative ligands.

A structural comparison between the VSG MiTat 1.2 N-terminus (PDB: 1VSG), MISP360, and the BARP model revealed they all share a similar height (~99.9 Å, ~93.0 Å and ~89.2 Å, respectively) (S18A Fig). However, when comparing the sequence length of their unstructured C-terminal domains (Fig 3G) with that of VSGs, it was clear that MISP360 and MISP380 have ~71% and ~34% longer C-termini; MISP440 has an equivalent length and MISP400/420 have a 60% shorter C-terminus. All BARP isoforms (except Tb927.9.15590) lack a C-terminal domain. By combining the crystal structure of MISP360 and the MISP C-terminal sequences, we generated structural models for all *T. brucei* MISP isoforms using I-TASSER [51–53] and IntFOLD [54] (S18B Fig). Similar to *Tc*HpHbR and VSG monomers, we predict that the MISP monomeric triple helical bundle is vertically oriented and, within the MCF surface context, is in close contact with neighboring mVSG homodimers. Since MISP360/380 have longer C-terminal domains than VSGs, and despite their lack of secondary structure, we predict that the MISP N-terminal domains (except for MISP400/420) could extend above the mVSG coat (Figs 3H and S18B). Consistent with our live immunostaining results (S14 Fig), these data suggest that despite being covered by a dense coat of mVSG homodimers, metacyclic trypomastigotes still allow room for the co-exposure of invariant GPI-anchored glycoproteins throughout the cell surface like BARP and MISP.

## MISP does not protect mice against a tsetse-transmitted *T. brucei* infection

Since we found MISP isoforms to be highly conserved, expressed on the metacyclic surface and susceptible to antibody binding in native conditions, we considered MISP as transmission-blocking vaccine candidates against *T. brucei*. To explore the experimental immunoprotective properties of MISP, the N-terminal ectodomain of MISP360 (Fig 4A) was

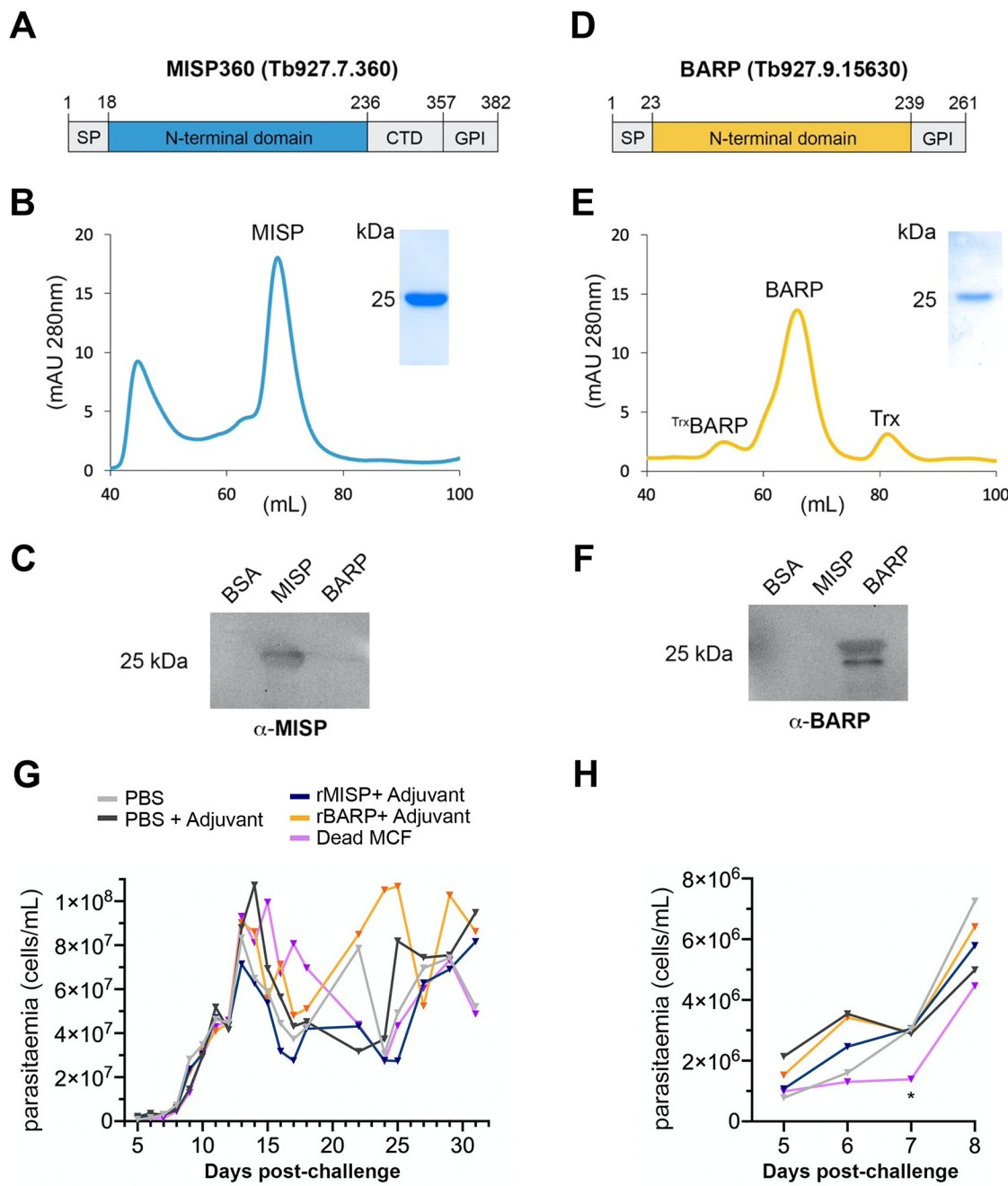

**Fig 4. Immunoprotective potential of MISP and BARP.** Representation of MISP (**A**) and BARP (**B**) recombinant proteins produced for mouse vaccination; sections in gray were not included; signal peptide (SP), C-terminus domain (CTD), GPI anchor peptide (GPI); numbers indicate position from Methionine 1. Size exclusion chromatography chromatograms of recombinant MISP (**C**) and BARP (**D**), indicating the purified peaks including the SDS-PAGE of the peak fraction as inset. Western blotting of BSA as control, recombinant MISP and BARP, probed with either anti-MISP polyclonal antibodies (**E**) or anti-BARP (**F**). (**G**) *T. brucei* parasitaemia development (parasites/mL blood) in mice either vaccinated with PBS only (gray), PBS and adjuvant (dark gray), recombinant MISP and adjuvant (blue), recombinant BARP and adjuvant (yellow) or dead metacyclic cells (purple) from 5 to 32 days post-challenge; dots indicate the mean of each group (*n* = 10 per group). (**H**), parasitemia levels from 5 to 8 days post-challenge highlighting the significant (*) reduction of parasitemia at day 7 post-challenge in mice vaccinated with dead metacyclic cells.

recombinantly produced in *E. coli*, purified by nickel affinity and size exclusion chromatography (Fig 4B–4C), depleted from endotoxins, and used to formulate a vaccine with glucopyranosyl lipid adjuvant aqueous formulation (GLA-AF) as adjuvant. In parallel, we also immunized mice with recombinant BARP (Fig 4D–4F) and dead whole *T. b. brucei* metacyclic cells. Ten BALB/c mice per group were immunized by subcutaneous injection and received two more boost doses after 10 and 20 days (Fig 4G–4H and S2 File). Ten days after the last booster, each mouse was bitten once by a single tsetse with a confirmed SG infection. Infection was followed up to 31 days post-challenge and parasitemia was regularly measured for each individual (Fig 4G). Animals vaccinated with either MISP or BARP developed an infection and did not show a significant reduction in parasitemia compared with the control groups (PBS and adjuvant only), at any point. Only mice immunized with dead metacyclic cells showed a significant reduction of parasitemia at 7 days post-challenge (Fig 4H), but this was restored to levels comparable to the controls the day after. Despite the immunized animals successfully developing an immune response against either antigen by generating high titers of specific antibodies as shown by immunoblotting (S2 File), vaccination did not prevent the development of a *T. b. brucei* infection.

## MISP is not essential for trypanosome colonization of the tsetse salivary glands

To gain more insight into the essentiality of MISP, we created a *T. b. brucei* transgenic cell line that allows the RNAi knockdown of all *misp* paralogs at once through a tetracycline-inducible RNAi stem loop system [55] targeting a conserved gene region. Although *misp* knockdown was successfully induced, both transcript and protein downregulation were only partial and did not lead to any conclusive phenotype either in PCF *in vitro* or within the tsetse salivary glands (S19 Fig). We then used a CRISPR-Cas9 strategy to knock out (KO) *misp* by targeting DNA breaks at both 5' and 3' sequences conserved in all paralogs, and replacement with drug selection markers (Fig 5A). An addback (AB) rescue cell line was created by inserting a tetracycline-inducible ectopic *misp360* gene for overexpression. As expected, due to the native expression pattern of *misp*, neither PCF cell line presented any growth or morphology phenotype *in vitro*. We then infected tsetse with either *misp* KO or AB cell lines and assessed infection rates at 30 dpi. All cell lines (i.e. KO, uninduced (AB-) and induced (AB+) addback) yielded normal midgut and proventricular infection rates, comparable to that of wild type cells. However, salivary gland infection rates of KO and AB- experienced a slight (although not statistically significant) reduction (Fig 5B) despite the observation that morphology and motility were apparently not affected, suggesting that MISP may have a minor but non-essential role in salivary gland infection establishment and/or maintenance. Lastly, *misp* KO cells re-expressing MISP360 (AB+ cells) failed to restore normal salivary gland infection rates. It remains to be determined whether this is due to the differential expression level compared to the endogenous gene and/or whether more than one isoform is necessary for complementation. Lack of MISP expression in KO cells was confirmed by immunostaining with anti-MISP antibody (Fig 5C), while expression of BARP and CRD remain comparable to the wild type and AB+ cells (S20 Fig).

## Discussion

Transmission of vector-borne pathogens usually requires insect saliva as a vehicle, but it is also accompanied by a series of soluble components from both the parasite and the insect. For instance, in the sand fly, *Leishmania* spp. parasites secrete abundant promastigote secretory gel and exosomes that are important virulence factors for transmission and establishment of the

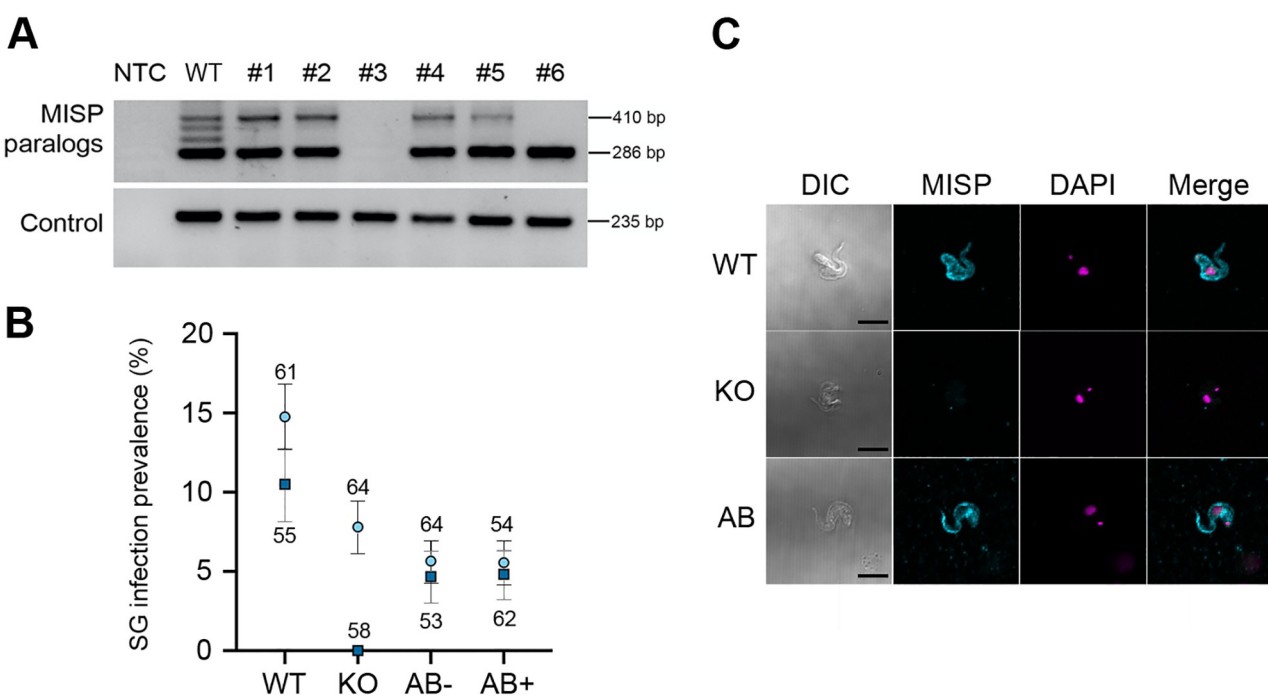

**Fig 5. Characterization of MISP null mutants.** (**A**) PCR genotyping of MISP KO clones (#1 to #6), wild type (WT), and negative template control (NTC) by amplification of MISP C-terminal domains (top gel); amplification of *tert* as control (bottom gel); relative molecular weight on the right (bp). (**B**) Infection prevalence (percentage) in tsetse salivary glands infected with either wild type (WT), MISP knock-out clone #3 (KO), MISP KO uninduced addback (AB-), and MISP KO induced addback (AB+); two biological replicates (replicate 1 (light blue circle); replicate 2 (dark blue square)); error bars indicate binomial ±SE of sample proportion (replicate 1(black), replicate 2 (gray)); sample population indicated next to each point. (**C**) Representative images of wild type (WT), MISP knock out (KO) and induced MISP addback (AB) immunostained with anti-MISP polyclonal antibodies (cyan), DAPI (magenta), merge, and differential interference contrast (DIC); scale bars = 5 μm.

parasite infection in the vertebrate host [56–58]. Here, we performed semi-quantitative prote-omics to analyze the composition of *T. b. brucei*-infected tsetse (*Glossina m. morsitans*) saliva with the aim of identifying potential soluble factors involved in parasite transmission. Along-side hits from *Glossina*, we found proteins from the bacterial symbiont *S. glossinidius*, which compared to naïve flies seem to be more abundant in saliva during a trypanosome infection, as shown by immunoblotting. This could be explained in part by a higher cell permeabilization or microbial dysbiosis led by the trypanosome infection, as *Sodalis* is capable of intracellular and extracellular growth within tsetse, including in the salivary glands [39]. An overall reduc-tion in the number of salivary proteins was confirmed in infected saliva as previously observed [32,34]. In addition, we identified 27 different *T. b. brucei* proteins in infected saliva, of which 62.9% were cytosolic and cytoskeletal. It remains unclear why trypanosome proteins are found in tsetse saliva and how they are released by the parasite. Their source could be dying parasites, and/or a more specific mechanism of protein secretion such as shedding of extracellular vesi-cles (EV) [59]. Indeed, it has been demonstrated that *T. brucei* BSFs secrete EVs that contain several virulence factors and proteins involved in parasite persistence within the host, includ-ing serum resistance-associated protein, VSGs, adenylate cyclase (GRESAG4), and GPI-PLC [60]. Notably, we showed that *T. brucei*-infected saliva is particularly enriched with trypanosome GPI-anchored surface proteins (37.1%); i.e. several VSGs, BARPs, and one (Tb927.7.360) belonging to a small family of hypothetical proteins, we herein named as meta-cyclic invariant surface proteins (MISP). Nevertheless, it remains to be determined whether

MISP, VSGs, BARP, and other proteins here found in tsetse saliva are part of EVs, secreted in soluble form, or both.

We identified and characterized the MISP family by studying gene and protein expression during parasite life cycle, cellular localization and accessibility to antibodies, protein tertiary structure, immunoprotective potential, and essentiality for parasite development in the tsetse vector. Most of these studies were conducted alongside BARP, a known family of surface proteins expressed by epimastigotes in SGs [21]. In our proteomic analyses, we detected at least four different BARP isoforms in infected saliva (Tb927.9.15520, Tb927.9.15530 and Tb927.9.15570), in agreement with a previous report [34]. MISP were previously included in the trypanosome surface phylome [40], within the Fam50 family that interestingly also includes BARP, and the *T. congolense* proteins GARP [61,62] and CESP [63]. Apart from *T. vivax*, genomes from most of the disease-relevant species of African trypanosomes such as *T. brucei* (including *T. b. gambiense* and *T. b. rhodesiense*), *T. congolense*, and *T. evansi* encode *misp* homolog genes. The preservation of MISP only across trypanosome species with full development in the tsetse vector suggests a conserved function and potential role in development or transmission.

We have confirmed that *misp* and *barp* gene expression is developmentally regulated in *T. b. brucei* SG stages as previously determined [36,37,64]. Our semi-quantitative RT-PCR method has allowed the detailed study of individual *misp* paralogs by exploiting the different number of repetitive elements at their C-termini. We found that *misp* up regulation is not homogeneous across isoforms, but instead biased towards a preferential expression for the shortest *misp-B* paralogs. Except for *misp-440* that seems to have a low but stable expression, *misp-A* transcripts are only detected during the late stages of parasite development in SG. It is unknown how these genes are expressed in such different ways despite being part of a gene array that is polycistronically-transcribed with >99.7% sequence conservation in UTR and intergenic sequences. Differences in homolog transcript levels, however, may not reflect changes in the respective protein abundance, which remain unknown as the polyclonal antibodies used in this study are expected to cross-react with all isoforms due to the overall high amino acid sequence conservation. Although proventricular trypanosomes produce large amounts of *misp* and *barp* transcripts [21], these are only translated into proteins in SG stages and localize evenly across the cell surface. While BARP expression peaks in SG epimastigotes and is gradually reduced during metacyclogenesis, MISP expression seems to increase with development and peaks in metacyclic forms, leading us to consider MISP as predominant metacyclic surface proteins (S21 Fig). Our observations disagree with a previous study [38] describing MISP (therein named SGE1) as proteins exclusively expressed by SG epimastigotes and absent in metacyclic forms, based on single-cell RNA-seq and immunostaining analyses. Such divergences may be attributed to several factors, including (1) use of different *T. b. brucei* strains and tsetse infection timing; (2) RNA-seq data not correlating with actual transcript translation like in PV-associated trypanosomes presenting high levels of MISP and BARP transcripts but not proteins; and (3) use of different polyclonal antibodies, particularly relevant when having to bind to scarcely accessible epitopes within the mVSG coat in metacyclic cells. Instead, we used confocal fluorescence microscopy and HRSEM to confirm the surface localization of MISP in metacyclic cells, and importantly, demonstrated that MISP epitopes are susceptible to antibody binding using live cells. Lastly, two more recent studies also support the expression of MISP in metacyclic cells using scRNA-seq [65], or a combination of transcriptomics and proteomics [66] (S3 File). Of importance, these studies were performed using either EATRO1125 [65] or Lister 427 (29:13 clone) parasite strains [66], which together with our MISP immunodetection in TSW196 (S15 Fig), strongly suggest that MISP expression in AnTat 1.1 metacyclic trypanosomes is not strain-specific. As for BARPs, previous observations

pointed at their expression being lost in MCFs [20,23], but we detected significant amounts of transcripts and protein levels in MCFs with variable localization (2C Fig). A similar effect was observed in PCFs where ectopic MISP localized to the flagellum, but then regained surface localization when cells differentiated into MCFs (where MISP are natively expressed).

Metacyclic forms are cells with an arrested cell cycle with low transcription activity. Therefore, detecting high levels of *misp* transcripts prompted us to hypothesize that these proteins may be abundant and play an important role in parasite development, transmission, or in establishing a mammalian host infection. To test this, we attempted to determine *misp* essentiality by knocking down transcripts from all paralogs using a single RNAi construct due to the high sequence conservation. The developmental expression of *misp* only in SG stages limited the selection process of clones in the culturable stages of the parasite (PCF). Although partial downregulation was successfully achieved in salivary gland trypanosomes, MISP could still be detected in MCFs making results inconclusive despite the absence of phenotype. We then knocked out all ten *misp* alleles using CRISPR-Cas9, which did not result in any apparent phenotype *in vitro*, and only produced a slight reduction in salivary gland infectivity, partially supporting the findings obtained by RNAi. Altogether, these data suggest that MISP are not essential for the development of *T. brucei* in the vector and may function during or after transmission to a mammalian host [26]. It remains to be determined whether MISP subfamilies have either distinct or complementary functions, as adding back a single *misp* did not rescue the slight infection phenotype observed, which may be simply due to loss of virulence from the extended culture time needed to generate the transgenic cell lines.

To gain more insight into the function of MISP, we solved the crystal structure of the ordered N-terminal domain of the MISP360 (Tb427.07.360). Structural analysis revealed a triple helical bundle structure that resembles other trypanosome surface proteins such as the *Tb/Tc*HpHbR [67], GARP [48] or a VSG monomer [47]. However, it is important to consider that mature MISP360 contains an additional C-terminal domain predicted to be highly disordered. In the absence of definitive structural data, we modelled the C-terminus, which indicated that it might serve as a long tether effectively elevating the N-terminal head group from the surface of the parasite membrane. This may partly explain the recognition of MISP by polyclonal antibodies shown by live immunostaining. It is tempting to speculate that this extension facilitates a biologically relevant interaction between the parasite and its environment such as promoting the acquisition of nutrients, serving as an adhesion or decoy, or modulating a response in the vertebrate host. The previously characterized *Tb*HpHbR lies partially within the VSG layer of the BSF and allows the trypanosome to acquire nutrients from the blood of the mammalian host. The mechanism by which it acquires nutrients is facilitated by a C-terminal extension that increases its relative distance from the membrane, thus making the ligand-binding site accessible by protruding above the VSG layer [45]. Since MCFs express mVSGs [68] alongside MISP, a C-terminal extension could support a similar role although the substantial length of the extension would likely render cellular uptake of a nutrient payload difficult. However, the unstructured extension is also suited to support an adhesion role for MISP, similar to the function described for the *Haemophilus* Cha adhesins [69]. Intriguingly, the only study that defined an adhesin function of a trypanosome surface protein (*T. congolense* CESP) is also part of the Fam50 family of proteins [63], thereby suggesting that MISP could also play a role in adhesion. Our structural analyses revealed the existence of a predominantly hydrophobic pocket in the apical end of MISP360, which we suggest may allow the coordination of a molecular partner, consistent with the putative role as an adhesin. Despite high sequence identity between isoforms, the residues forming and surrounding the pocket lacked sequence conservation and resulted in a distorted cavity between the homolog models that may reflect the capacity to coordinate a structurally diverse set of ligands, potentially from different hosts. Furthermore,

the homologs differ in the length of the C-terminal region, originating from the variable number of 26-residues motifs. This region could offer the parasite a robust mechanism to engage a variety of cell types or provide a sequential binding effect to enable tighter adhesion. It is noteworthy that these putative unstructured C-terminal regions contain lysine and arginine residues, which are inherently susceptible to proteolysis consistent with predictions using the PROSPER software [70] (S22 Fig). The predicted proteolytic susceptibility may serve a strategic function in enabling the parasite to release the molecular interactions anchoring it to the tsetse salivary glands and thereby promote transmission. It has been assumed that the MCF surface is structurally and functionally equivalent to that of BSFs as they show the same thick VSG (electron-dense) layer in TEM sections [12,24,71]. This implies that mVSGs also form a dense macromolecular barrier that possibly masks MCF invariant surface proteins from the host's immune system. Most of the characterized BSF invariant surface proteins (e.g., ISG65, ISG75, ESAG4 and hexose transporters), which are found on either the flagellar or cell surface, are attached via transmembrane domains [8,11]. Two exceptions are the transferrin [9] and the HpHbR receptors [10], which are GPI-anchored and exclusively localize in the flagellar pocket. This restricted localization has been proposed to be due partly to the presence of only one GPI anchor molecule, differing from VSGs, which form homodimers (i.e. two GPI anchors) that allow their homogeneous surface expression [72]. In contrast, our structural and immunostaining evidence shows that MISP appear to be displayed on the entire cell surface as monomer. This may suggest that, unlike BSFs, MCFs allow the homogeneous surface expression of monomeric GPI-anchored invariant proteins, although we cannot rule out the possibility of MISP multimerizing *in vivo*. This clearly differs when MISP is ectopically expressed in BSF or PCF, in which the proteins localize at the flagellar pocket and/or the flagellar membrane. It remains to be determined the mode of association to the metacyclic membrane of another small family invariable surface proteins known as Fam10 [40], as these molecules lack identifiable hydrophobic transmembrane domain or GPI-signal addition site in the C-terminus.

Due to its surface exposure, antigenicity, and conservation, we considered MISP as potential vaccine candidates. Our immunization experiment failed to provide protection against a tsetse challenge, as it also failed against *T. b. brucei* needle challenge in a previous study [38]. However, in neither study was the MISP immunogen designed for optimal performance; improvements could include: 1) using an alternative expression system (e.g. *Leishmania tarantolae*) to conserve all post-translational modifications; 2) immunizing with full length protein; or 3) testing different adjuvants that elicit a protective immune response against a trypanosome infection.

Our proteomic analyses identified peptides from several VSG proteins. The identification of mVSG peptides in infected saliva is not surprising given that infected SGs contain thousands of quiescent metacyclic cells, which may release mVSGs into the SG lumen. mVSG release may occur partly by the action of the parasite GPI-PLC, which is highly expressed in MCFs [23,73] and/or by the action of tsetse salivary proteases. Interestingly, among the VSG species detected, only one corresponds to a canonical mVSG (mVAT4), which has not been previously detected as protein, but its metacyclic expression site (MES) has been well characterized [74]. Notably, a similar study detected mVAT5 in tsetse saliva infected, but from a different *T. brucei* strain [34]. Although it has been reported that BSFs may express mVSGs from bloodstream expression site (BES) at low levels [75], no study has described the expression of BSF VSG proteins in MCFs. This supports the assumption that, although recombination may occur from MES to BES, it does not appear to occur in the opposite direction since MES lack the 70 bp repeats upstream of the mVSG gene. However, we identified peptides found in at least four BSF VSGs, including MiTat 1.2 (identified by three unique peptides). This suggests that either (1) MCF

could have active BES (unlikely since MCFs do not appear to express ESAG proteins [73]) or 2) there is recombination at the MES replacing the *mvsg* by a *vsg* gene, or 3) there is recombination between *mvsg* and *vsg* genes leading to the formation of mosaics. In addition, we cannot rule out that the undetermined VSG repertoire of the TSW196 strain [76] (used in this study) could differ from other reference strains [77]. Importantly, previous studies have found similar results although this aspect has been overlooked [34,36,64,73] (S6 Table). For example, at least two BSF VSGs were identified in saliva from flies infected with *T. b. brucei* EATRO 1125 [34]; one BSF *vsg* was found to be highly transcribed in a *T. b. brucei* RUMP503 SG infection [64]; and 15 BSF *vsg*s had similar transcription levels than *mvsg*s (and a few were detected as protein) in *T. b. brucei* Lister 427 MCF overexpressing the RNA-binding protein 6 [73].

In summary, not only does the *T. brucei*-infected tsetse saliva contain infective MCFs, but also a cocktail of *G. morsitans* salivary proteins, bacterial and trypanosome soluble factors. How this complex composition modulates parasite transmission is a fascinating question yet to be addressed. Among the detected parasite proteins in infected saliva, MISP represent a family of immunogenic, invariant surface proteins that are exposed on the cell surface of the mammalian-infectious metacyclic stage of *T. brucei*. Based on their external exposure and sequence conservation, we considered MISP as potential vaccine candidates against *T. brucei* spp-caused trypanosomiasis, in a similar way to the malaria vaccine that targets the sporozoite CSP protein [78]. Lastly, it could be advantageous to use MISP as a xenodiagnosis marker in the field for the identification of disease-transmissible tsetse. Having a parasite marker that discerns between a mature tsetse salivary gland infection (disease vector) and trypanosomes in the midgut (potential non-vectors), would provide a key tool for accurately assessing both vector incrimination and the prevalence of trypanosomiasis in the field.

## Methods

### Ethics statement

All applicable international, national, and institutional guidelines for the care and use of animals were followed. Specifically, the experimental use of the BALB/c female mice for the immunization and tsetse fly-mediated trypanosome infection was approved by the Institute of Tropical Medicine Antwerp (ITM) Animal Ethics Committee Clearance under licence No. ECD VPU2018-1.

### *T. b. brucei* cell culture

*Trypanosoma brucei brucei* strain AnTat 1.1 90:13 PCF [55,79] were cultured in SDM-79 medium supplemented with 10% FBS, haemin, 2 mM L-glutamine, Glutamax (Gibco), 15 µg/mL G418 (InvivoGen) and 50 µg/mL hygromycin B (InvivoGen), at 27˚C and 5% $CO_2$. Cells were typically passaged every 2 days. Transgenic AnTat 1.1 90:13 PCF cell lines were cultured in the presence of the selection marker drugs puromycin 1 µg/mL or zeocin 10 µg/mL (Invivo-Gen). Cells requiring the activation of the transgene were cultured in the presence of fresh 1µg/mL doxycycline (in 70% ethanol) or the same volume of 70% ethanol for negative controls. *T. b. brucei* AnTat 1.1 90:13 BSF were cultured in HMI-9 medium supplemented with 10% FBS, 10% Serum Plus (Sigma) and GlutaMax (Gibco), 2.5 µg/mL G418 and 4 µg/mL hygromycin B at 37˚C and 5% $CO_2$.

### Tsetse fly infection and dissection

*G. m. morsitans* were obtained from the tsetse insectary at the Liverpool School of Tropical Medicine, where they were maintained at 26˚C (+/-1˚C) and 68–78% relative humidity. Flies

were fed for 12 minutes every second day on sterile, defibrinated horse blood (TCS Biosciences) through a silicone membrane feeding system. Teneral (24h post emergence, unfed) males were infected by combining *T. b. brucei* TSW196-infected rat blood with fresh, sterile, defibrinated horse blood at a density of $10^6$ cells/mL (for saliva proteomics, S1 Fig). Alternatively, flies were infected by mixing either *T. b. brucei* AnTat 1.1 90:13 [55] or AnTat 1.1 Cas9 [80] PCF cells with washed, defibrinated horse blood at a density of $5x10^5$ cells/mL. Uninfected control flies were fed on the same blood (free of trypanosomes) for the same amount of time. Any experimental flies that had not fed were removed the following day. Experimental flies were maintained for 4 weeks, feeding on normal defibrinated horse blood every 2 days. Alternatively, flies infected with mutant trypanosomes requiring tetracycline activation of the transgene were fed with normal defibrinated horse blood supplemented with fresh 2 μg/mL doxycycline (in 70% ethanol). Bloodmeals for control flies were supplemented with the same volume of 70% ethanol only. Flies were dissected at 28 to 30 days post infection, 72 hours after receiving the last bloodmeal. For saliva proteomics, based on normal *T. b. brucei* TSW196 salivary gland infection rates (~15%) and yields (~$10^4$ trypanosomes/gland pair), 1,500 naïve and 1,500 *T. b. brucei*-infected tsetse were dissected (3 biological replicates of ~500 male flies per group). Salivary glands were collected in sterile ice-cold PBS and centrifuged at 12,000 rpm for 5 minutes at 4˚C to collect the resultant saliva supernatant. SG and saliva were then separately stored at -80˚C until further analyses. Alternatively, the tsetse SG, PV and MG were dissected out (in this order to prevent trypanosome cross-contamination) in sterile ice-cold PBS and *T. brucei* cells were isolated, washed in PBS by centrifugation at 2,000 rpm for 5 minutes at 4˚C and either stored at -80˚C or directly used for immunostaining.

## High-resolution nLC-MS/MS proteomic analysis

Saliva samples from naïve and *T. brucei*-infected tsetse were lyophilized and sent to the Dundee University Fingerprints Proteomics Facility for in-solution trypsin digests and identification by mass spectrometry. Briefly, lyophilized saliva samples (10–100 μg each) were redissolved in 200 μL 0.4 M $NH_4HCO_3$ containing 8 M urea, and disulfide bonds were reduced with 5 mM dithiothreitol for 15 min at 50˚C. Cysteine residues were alkylated with freshly prepared 10 mM iodoacetamide for 30 min at room temperature (RT), protected from light. Proteins were digested with 20 μg sequencing-grade trypsin (Sigma-Aldrich) for 24 h at 37˚C. The reaction was terminated by adding 10 μL formic acid. The resulting peptides were desalted in a C18-SPE cartridge (1 mL, Discovery DSC-18, Supelco, Sigma-Aldrich), and analyzed by nano-liquid chromatography-high-resolution tandem mass spectrometry (nLC-HR-MS/MS) using an Ultimate 3000 RSLC nano coupled to LTQ-Orbitrap Velos Pro MS (Thermo Fisher Scientific) and filtered with an estimated false-discovery rate (FDR) of 1%. The PD settings were as follows: HCD MS/MS; cysteine carboxyamidomethylation as fixed modification; methionine oxidation and lysine acetylation as variable modifications; fully tryptic peptides only; up to two missed cleavages; parent ion mass tolerance of 10 ppm (monoisotopic); and fragment mass tolerance of 0.6 Da (in SEQUEST) and 0.02 Da (in PD 2.1.1.21) (monoisotopic). Tandem MS/MS spectra were searched against the following databases: a combined protein database of *T. b. brucei*, *T. b. gambiense* and *T. b. rhodesiense* (downloaded in October 2017 from TriTrypDB; http://www.tritrypdb.org/ in FASTA format, release 34, 18,729 entries, and from UniProtKB; http://www.uniprot.org/; release 2017_10, 25,450 entries), and *G. m. morsitans* (downloaded in May 2017 from VectorBase; https://www.vectorbase.org/; version 1.7, 12,494 entries). Common contaminant sequences (MaxQuant) were included in this combined database. The resulting PD dataset was further processed through Scaffold Q+ 4.8.2 and perSPECtives (Proteome Software, Portland, OR). A protein threshold of 95%, peptide threshold of 95%, and a

minimum number of one unique peptide was used for identification of proteins. Manual inspection of the MS/MS spectrum was always performed for identification of any protein based on a single peptide. Using BLAST2GO, proteins were BLAST searched against the non-redundant NCBI protein database and mapped and categorized according to gene ontology terms (GO). *Sodalis glossinidius* proteins were identified using the Liverpool School of Tropical Medicine in-house MASCOT MS/MS ion search using a protein database from UniProtKB generated from the latest re-annotated coding sequences in *S. glossinidius* [81], using the following settings: two missed trypsin cleavages, same peptide modifications described above, peptide tolerance of 1.2 Da, MS/MS tolerance of 1.2 Da and peptide charges set to +1, +2 and +3. The mass spectrometry proteomics data have been deposited to the ProteomeXchange Consortium via the PRIDE (www.ebi.ac.uk/pride/) partner repository with the dataset identifier PXD039684.

## Bioinformatic analyses

All trypanosome sequences were obtained from TriTrypDB (www.tritrypdb.org) [82] or UniProt (www.uniprot.org) [83]. *G. m. morsitans* sequences were obtained from VectorBase (www.vectorbase.org) [84]. Sequence analyses and multiple alignments were done using Clustal Omega [85] and Geneious R9 (Biomatters Ltd). Alignment figures were made in Jalview [86]. Phylogenetic trees were generated using PhyML 3.1 [41] under default conditions with a set bootstrap of 500. Signal peptide, GPI-anchor signal peptide, and transmembrane domain predictions were determined using the servers SignalP 4.1 [87], PredGPI [88] and TMHMM 2.0 [89,90], respectively, under default settings. *N*-glycosylation predictions were run using the NetNGlyc 1.0 server (http://www.cbs.dtu.dk/services/NetNGlyc). MISP 3D protein modelling was performed using I-TASSER [51–53] with the crystal structure of MISP360 as a structural template and multiple sequence alignments (Clustal Omega) with the query sequences, or with IntFOLD [54]. Structural homology searches were performed using DALI [91] and the conservation models using ConSurf [49,50].

## PCR and semi-quantitative RT-PCR

Total RNA from cultured and tsetse-isolated trypanosome samples was extracted using the GeneJet RNA extraction kit (Thermo Scientific) following the manufacturer's protocol. Extracted RNA was further treated with the TURBO DNAse kit (Ambion) to remove residual genomic DNA following the manufacturer's instructions. Variable amounts of RNA were reverse-transcribed into cDNA at 52˚C for 10 minutes using SuperScript IV (Invitrogen) and oligo-dT$_{20}$ primer, inactivated at 80˚C for 10 minutes and treated with *E. coli* RNAse H at 37˚C for 20 minutes. The resultant RNA-free cDNA was used in a downstream PCR using the Taq 2x Mastermix (New England Biolabs) and the primer pairs (400 nM) 5'-GCCGAAAGGA CGGCAGAGACG-3' and 5'-GTGTATCCTCCTATAGATT CTGCATAGC-3' for the detection of individual *misp* homologs (amplicons of 410 bp, 332 bp, 254 bp and 176 bp for *misp360*, *misp380*, *misp440* and *misp400/420*, respectively); 5'-GGCTACTTTTGACAGTGTT ATG-3' and 5'-CTTCCTTCGCTTTCGTAACG-3' to detect all *barp* homologs (270 bp amplicon). To determine *misp* knockdown levels in *misp^{RNAi}* cells, the primers 5'-GATGCCTCTAA AGCGAGAGAAGG-3' and 5'-GTACTGGCGGCACTATTGTCCTC-3' were used (263 bp amplicon). Expression values were normalized against the expression of *tert* [92,93] using the primers 5'-AACAGTAAGTCCCGTGTGG-3' and 5'-CAAAGTCGTACTTCAACATCAG-3' (233 bp amplicon). For all RT-PCR reactions, we ran a non-template control, a control for DNAse treatment, for reverse-transcription and a positive control with 1 ng genomic DNA for standardization. PCR conditions were as follows: 95˚ for 2 minutes, followed by 30 or 35 cycles

of 95˚C for 30 seconds, annealing for 30 seconds at variable temperatures, and elongation at 68˚C for 30 seconds. A last elongation step at 68˚ was performed for 2 minutes. The cycle number was set at the exponential phase of the reaction. All RT-PCR products were mixed with sample buffer and run in a 2% agarose gel at 100V for 50 minutes. Gels were stained with SYBR Safe (Invitrogen), imaged using a Bio-Rad Gel Doc EZ system and analyzed using the ImageLab software. Imaging conditions were kept identical for all measurements for comparison.

### Generation of anti-MISP and anti-BARP antibodies

Anti-MISP polyclonal antibody was generated by immunization of a single rabbit (Davids Biotechnologie GmbH) with 3 courses of 0.3 mg of recombinant MISP380 (full Hisx6-tagged protein lacking N- and GPI-signal peptides produced in Rosetta 2 (DE3) *E. coli* and purified by nickel affinity followed by size exclusion chromatography), and further purification of IgG from antiserum (63 days post immunization bleed) by protein A and affinity (MISP) chromatography, and posterior exclusion chromatography using DE3 *E. coli* lysate). Anti-BARP polyclonal antibody was generated by immunization of a single rabbit (Eurogentec) with BSA conjugated to the epitope peptide sequence CKVQAEEAVELAESK (fully conserved across all BARP isoforms) and further purification of specific antibodies by protein A and affinity chromatography. We tested the specificity of either antibody using western blotting by probing recombinant MISP, BARP and soluble VSG221 (S23 Fig).

### Immunofluorescence assay and confocal laser scanning microscopy

Cultured or tsetse-derived *T. b. brucei* cells were washed in PBS and fixed in fresh 4% paraformaldehyde (PFA) for 30 minutes at RT. Cells were then allowed to settle onto poly-L-lysine coated slides for 20 minutes in a humidity chamber. Alternatively, cells were first air-dried on a poly-L-lysine slide, fixed in 100% methanol for 10 minutes at -20˚C and rehydrated in PBS after air-drying. PFA-fixed cells were then either permeabilized or not in 0.1% Triton X-100 for 10 minutes at RT. They were then blocked in vPBS (PBS containing 45.9 mM sucrose and 10 mM glucose) 20% FBS for 1 hour, washed in PBS and incubated for 1 hour at RT with the primary antibody solution; 1:200 (anti-MISP (this study), anti-BARP (this study), anti-HA (ThermoFisher), 1:50 (anti-CRD (Glyko)), 1:100 (anti-PFR (a generous gift from Prof Helen Price, Keele University), 1:800 (anti-EP (Cedar Lane)) dilutions in vPBS 20% FBS. After further washing, they were incubated for 1 hour at RT with the secondary antibody solution (1:500 goat-anti-mouse/rabbit Alexa Fluor 488/555; ThermoFisher). Cells were then incubated with 300 ng/mL 4',6-diamidino-2-phenylindole (DAPI) for 10 minutes, washed and mounted in Slowfade Diamond mounting oil (Life Technologies). Slides were imaged and analyzed using the confocal laser scanning microscope Zeiss LSM-880 and the Zeiss Zen software. For mean intensity fluorescence measurements, images were taken conserving laser power, pinhole aperture and digital gain settings. Images were further analyzed using Zen and ImageJ.

### High-Resolution Scanning Electron Microscopy (HRSEM)

Both *T. brucei* PCF and MCF cells were fixed in 4% PFA 0.02% glutaraldehyde in 0.1M phosphate buffer (PA) for 30 minutes at RT, washed in PBS and attached to poly-L-lysine coverslips. Cells were blocked in 1% BSA PBS for 30 minutes and incubated with primary anti-MISP antibodies (1:50 in blocking solution) for 1 hour at RT. After washing, cells were incubated with secondary IgG goat anti-rabbit conjugated to 10nm gold nanoparticles (1:50 in blocking solution) for 1 hour at RT. Cells were washed and fixed in 2.5% glutaraldehyde 0.1M PA for 30 minutes at RT, washed in water and dehydrated in 30, 50, 70, 90 and 100% ethanol

solutions progressively, for 10 minutes each. Cells were finally dried using a Tousimis automatic critical point dryer and coated in 4nm palladium using an Agar Sputter Coater. Samples were imaged in a merlin VP compact high-resolution scanning electron microscope (Zeiss) equipped with a 3View stage (Ametek) and an OnPoint back-scattered electron detector (Ametek). The samples were imaged under vacuum with Nitrogen gas. The microscope conditions were as follows: aperture 20μm, working distance 5-6mm, Z height 47mm, EHT (accelerating voltage) 10kv, BSD of very high gain, brightness 46.9% and contrast 87.7%.

## Generation of [HA-eGFP]MISP360 and *misp[RNAi]* PCF transgenic cell lines

To create a cell line expressing ectopic MISP360 tagged with an HA-epitope and eGFP at N-terminus ([HA-eGFP]MISP360), the nucleotide sequence coding for the mature Tb427.07.360 protein lacking the signal peptide (Asp[18] to Phe[382]) was obtained by PCR (Q5 high-fidelity, New England Biolabs) from *T. b. brucei* (strain AnTat 1.1 90:13) genomic DNA using the following primer sequences: 5'-GCG<u>TCTAGA</u>GACTCCATAATTGAGGAAGG-3' and 5'-GCG<u>AGATCT</u>TTAAAAATGTGCGGCAGC-3' with *XbaI* and *BglII* restriction enzyme targets (underlined), respectively. The Tb927.7.360 signal peptide (Met[1] to Ala[17]) coding sequence was separately obtained by annealing the oligonucleotides 5'-CGC<u>AAGCTT</u>ATGACTAGCCCGTTTT TTGTGCTTGCTTGCTATACTTACGTATGTCACCGCC<u>AAGCTT</u>CGC-3' and 5'- GCG<u>AA GCTT</u>GGCGGTGACATACGTAAGTATAGCAAG

CAAGCACAAAAAACGGGCAGTCAT<u>AAGCTT</u>GCG-3' with *HindIII* at both ends, and the HA epitope coding sequence was obtained by annealing the oligonucleotides 5'-CGC<u>AAG CTT</u>TATCCATATGACGTGCCGGATTATGCG<u>ACTAGT</u>CGC-3' and 5'-GCG<u>ACTAGT</u>CG CATAATCCGGCACGTCATATGGATA<u>AAGCTT</u>GCG-3' with *HindIII* and *SpeI*. The pDEX-577 plasmid [94] for the tetracycline-inducible ectopic gene expression under a T7 promoter, was digested with *HindIII* and *SpeI* and ligated with *HindIII/SpeI*-digested HA sequence. The resultant plasmid was digested with *HindIII*, dephosphorylated with recombinant shrimp alkaline phosphatase and the *HindIII*-digested signal peptide coding sequence was inserted. The construct was then digested with *XbaI* and *BamHI* to ligate the *Tb927.7.360* (lacking the signal peptide) coding sequence digested with *XbaI* and *BglII*. The resultant construct was linearized using *NotI* high-fidelity (New England Biolabs) and transfected into *T. b. brucei* (strain AnTat 1.1 90:13) PCF using an Amaxa 4D nucleofector, program FI-115 in Tb-BSF buffer [95]. Clonal transfectant lines were selected with 10 μg/mL zeocin. To create a cell line for the RNAi knockdown of *misp* (*misp[RNAi]*), a short unique 494 bp sequence highly conserved in all *misp* homologs was manually selected and obtained by PCR (Q5 high-fidelity NEB) using the primers 5'-CTACCACTACAGCAATTCGG-3' and 5'- GCCTCTCGCAATTC GTCTTG-3' with the restriction target pairs *XhoI/BamHI* or *NdeI/HindIII* respectively. By digesting with the corresponding restriction enzymes, the two amplicons were sequentially inserted in sense and antisense directions into the plasmid pALC14 [55] for a tetracycline-inducible transcription of a short hairpin double-stranded RNA. The plasmid was linearized (*NotI* high-fidelity) and transfected into AnTat 1.1 90:13 PCF as described above, and clonal transfectant cell lines were selected with 1 μg/mL puromycin. Transgene insertions were checked by PCR.

## Generation of MISP knockout and addback cell lines using CRISPR/Cas9

All ten *misp* alleles (5 paralogue genes) were knocked out in two consecutive rounds of CRISPR/Cas9 in AnTat 1.1 Cas9 as previously described [80], using the LeishGEdit toolkit [96], except for transfections being carried out in PCF cells obtained by *in vitro* differentiation from cultured BSF in DTM medium containing 6mM cis-aconitate at 27°C. Briefly, dsDNA

encoding for two different sgRNA targeting universal sequences upstream and downstream the coding sequence of all *misp* paralogs, were produced by PCR and co-transfected as described above with two donor DNA, consisting of either neomycin or hygromycin resistance genes flanked by homology regions found beyond the CRISPR-targeted sites. The addback cell line was generated by complementing the MISP KO clone 3 cell line with a pDEX377 plasmid containing the *misp360* gene for its tetracycline-inducible ectopic expression.

## Expression and purification of recombinant MISP360 and BARP for structural analyses

The nucleotide sequence encoding *Tb*427.07.360 (MISP360) (Gly$^{24}$ to Ala$^{234}$) and BARP ABB49055.1 (Glu$^{23}$ to Gly$^{238}$) were synthesized by GenScript and codon-optimized for *E. coli*. The constructs were sub-cloned into a modified pET32a vector (Novagen) containing N-terminal thioredoxin (Trx) and hexa-histidine (His$_6$) tags separated from the gene of interest by a TEV protease site. Sequencing confirmed that no mutations were introduced. *E. coli* Rosetta-gami 2 (DE3) cells (Novagen) were transformed with the plasmid and grown in autoinduction medium (Invitrogen) from a 5% inoculum. Following four hours of growth at 37˚C and 64 hours at 16˚C, cells were harvested, and the pellet re-suspended in 20 mM HEPES pH 8.3, 1 M NaCl, 30 mM imidazole and 5mM β-mercaptoethanol. Cells in suspension were lysed using a French Press, insoluble material was removed by centrifugation, and soluble fraction allowed to batch-bind with Ni-agarose beads at 4˚C for 1 hour. Recombinant MISP360 and BARP were eluted with the same buffer as before supplemented with 250 mM imidazole, and fractions were analyzed by SDS-PAGE and pooled based on purity. The Trx-His$_6$ tag was removed by TEV cleavage overnight at 291 K, and the proteins were further purified by size exclusion chromatography (SEC) (Superdex 16/60 200) in HEPES buffered saline (HBS) with 1mM DTT. Protein concentration was determined by absorbance at 280 nm with calculated extinction coefficients of 10220 M$^{-1}$ cm$^{-1}$ (MISP360), and 12740 M$^{-1}$ cm$^{-1}$ (BARP).

## Crystallization and data collection of MISP360

Crystals of purified MISP360 were initially identified in the Index screen (Molecular Dimensions) using the sitting drop method at 18˚C. The final drops consisted of 0.3 μL of MISP360 at 20 mg ml$^{-1}$ with 0.3 μL of reservoir solution and were equilibrated against 60 μL of reservoir solution. Diffraction-quality crystals grew in 48 hours in 25% PEG, 1500. A single crystal was looped, cryopreserved in 12.5% glycerol for 20 seconds, and flash-cooled to 100 K directly in the cryostream. Diffraction data were collected on beamline 9–2 at Stanford Synchrotron Radiation Laboratory (SSRL) at a wavelength of 0.9795 Å.

## MISP360 structure data processing, solution, and refinement

Diffraction data to 1.82 Å resolution were processed using Imosflm [97] and Scala [98] in the CCP4 suite of programs [99]. Initial phases were obtained by molecular replacement using PHASER [100] with one copy of GARP [48] (PDB 2Y44; 15% identity over 210 residues). Solvent molecules were selected using COOT [101] and refinement carried out using Phenix Refine [102]. The overall structure of MISP360 was refined to an R$_{free}$ of 20.43%. Stereo-chemical analysis performed with PROCHECK and SFCHECK in CCP4 [99] showed excellent stereochemistry with more than 99% of the residues in the favored conformations and no residues modelled in disallowed orientations of the Ramachandran plot. Overall, 5% of the reflections were set aside for calculation of R$_{free}$. Data collection and refinement statistics are presented in S7 Table. The atomic coordinates and structure factors have been deposited in the Protein Data Bank under the code 5VTL.

## *In silico* homology modelling of BARP

For modelling BARP, the crystal structure of GARP (PDB: 2Y44) and MISP360 were used as the templates, with 24 and 28% identity, respectively. Using Modeller 9v18 [43], 10 models of BARP were generated and the best model was chosen based on the low value of normalised discrete optimized protein energy (DOPE). The assessment of the final model was carried out with Ramachandran statistics [103], QMEAN [104], and ProSA [105].

## SDS-PAGE and western blotting

Trypanosome lysates were prepared by pelleting $10^6$ cells and heat-denaturing (98˚C for 10 minutes) in Laemmli buffer. Samples were resolved on a 12.5% SDS-PAGE gel using the Bio-Rad mini-Protean system at 50V for 30 minutes followed by 2 hours at 120V. Proteins were transferred from the gel to a polyvinylidene fluoride (PVDF) membrane. After transfer at 90V for 1 hour, membranes were blocked (PBS-Tween 0.5% v/v, 5% w/v skim milk powder) for 1 hour at RT and incubated with primary anti-HA (1:10,000; Thermofisher) or anti-*Sodalis* Hsp60 mAb 1H1 [106] (mouse anti-symbiont GroEL; 1:10 dilution) overnight at 4˚C. The membranes were then washed in PBS-Tween (0.5% v/v) and probed at RT for 1 hour with secondary goat-anti-rabbit or goat-anti-mouse IgG horseradish peroxidase-conjugated antibodies (1:25,000 dilution; Thermofisher). The membranes were washed and probed with Super Signal West Dura substrate (Thermo Scientific) and the signal transferred onto X-ray Amersham Hyperfilm (GE Healthcare). Membranes were lastly stained in 0.2% (w/v) nigrosine in PBS for 15 minutes for loading control. Alternatively, recombinant protein samples (S23 Fig) were denatured in LDS sample buffer with DTT reducing agent for 10 minutes at 70˚C, run in pre-cast 4–12% Bolt SDS-PAGE gels (Thermofisher) in MES buffer, dry-transferred onto PVDF membranes, blocked with 1% BSA in TBS, and incubated with primary antibodies for 1 hour at RT (anti-MISP 1:2,000; anti-BARP 1:2,000; anti-VSG 1:5,000). After washing, membranes were incubated with secondary Alexa Fluor Plus 800 anti-rabbit IgG antibodies (1:10,000, Thermofisher) for 1 hour at RT before imaging the membrane using an Oddysey Fc Imager.

## MISP and BARP vaccine formulation, mouse immunization and tsetse challenge

For immunization, the N-terminal ectodomain of MISP360 (Asp$^{18}$-Glu$^{237}$) was recombinantly expressed in *E. coli* Rosetta 2 (BL21) cells (Merk), fused to a 6xHis-tag at C-terminus. The recombinant BARP used in immunization was identical to that used for crystallization (see above). Recombinant proteins were purified through a Ni-NTA column and SEC-FPLC and depleted from endotoxins using the Pierce high-capacity endotoxin removal resin (Thermo Scientific). Endotoxin levels were then determined using the Pierce Chromogenic endotoxin quant kit (Thermo Scientific). Vaccines were formulated in liposomal-monophosphoryl-Lipid A (L-MPLA) as previously described [107,108]. Briefly, 1,2-dimyristoyl-*sn*-glycero-3-phospho-choline, cholesterol, and 1,2-dimyristoyl-*sn*-glycero-3-phospho-rac-(1-glycerol) (Avanti Polar Lipids, Alabaster, AL) were dissolved in freshly distilled chloroform stabilized with 0.75% ethanol (9:7.5:1 molar ratio) along with MPLA (Sigma-Aldrich, St. Louis, MO) for a final concentration of 100 μg/mL. Liposomes were dried under nitrogen stream and placed overnight in a desiccator containing silica-gel desiccant beads to obtain a white pellet (Thermo Fisher Scientific), and then vortexed to resuspend at 10 μg/100 μL Lipid A per dose using sterile PBS (pH 7.4) containing either 5 μg of rMISP360, 5 μg of rBARP, $10^5$ whole frozen metacyclics *T. b. brucei* Antat 1.1 in PBS, or PBS alone until completely dissolved. Vaccines were filtered (0.2-μm) before immunization and administered to groups of ten BALB/c 6–8 weeks old female mice

three times at days 0, 10 and 21 by sub-cutaneous injection at the base of the tail. Pre-immune sera were collected for each individual 5 days before the first vaccine was delivered. Mice were then anaesthetized with ketamine (100mg/kg) and xylazine (20mg/kg) and challenged 12 days after the last booster (day 33) with the bite of tsetse flies pre-selected for positive *T. b. brucei* AnTat 1.1 salivary gland infections, by allowing individual tsetse to bite on the mouse ear dermis. Parasitaemia development was monitored by microscopic analysis using Uriglass disposable slides (Menarini Diagnostics) from day 4 post-challenge onwards. The tsetse flies used for this challenge were infected by providing a first blood meal after emergence (<48 hours) containing $10^6$/mL *T. b. brucei* AnTat 1.1 pleiomorphic bloodstream forms in defibrinated horse blood supplemented with 10 mM reduced L-glutathione to increase trypanosome infection establishment. Flies were then maintained by feeding every 2–3 days on uninfected defibrinated horse blood. Salivary gland infected flies were pre-selected by induced probing on pre-warmed glass slides that were microscopically examined for the presence of metacyclic trypanosomes. To confirm the efficacy of vaccines in raising a specific IgG response against the antigens, we performed western blotting on recombinant MISP, BARP and metacyclic cell lysates probing with mouse antisera collected at day 30 post-vaccination (3 days pre-challenge). Briefly, samples were denatured in sample buffer and run in 10% SDS-PAGE gels, transferred onto nitrocellulose membranes, blocking in TBS-T 5% skim milk blocking buffer and probed with mouse anti-serum (1:200 dilution in blocking buffer) during 1 hour at RT. Membranes were then probed with anti-mouse IgG conjugated with HRP (1:20,000 dilution in blocking buffer) at 1 hour at RT, and antibodies detected using TMB-membrane peroxidase substrate (Seracare).

## Supporting information

**S1 Fig. Workflow for the analysis of *T. b. brucei*-infected tsetse saliva.** For each biological replicate, saliva samples from naïve flies were also collected and processed in parallel.
(TIF)

**S2 Fig. Chemiluminescent Western blot analysis of *G. m. morsitans* naïve and *T. b. brucei*-infected saliva.** Mouse anti-*Sodalis* monoclonal antibody 1H1 recognizes a heat shock protein (Hsp60) produced by *Sodalis glossinidius*. Lanes 1 and 2 represent film exposed for 10 seconds. Lanes 3 and 4 show the nigrosine-stained PVDF membrane used for blotting to visualize the protein loading in each lane. Naïve saliva (N). Infected saliva (I).
(TIF)

**S3 Fig. Multiple protein sequence alignments of MISP. A**, Multiple protein alignment of all MISPs (*T. congolense* MISP excluded). Residues colored based on identity conservation from black (maximum) to red (minimum). Alignment made using a Clustal Omega BLOSUM62 matrix with a gap open cost 10 and a gap extension cost of 0.1, using Geneious R9. Domains highlighted: ER signal peptide (green), potential *N*-glycosylation sites (yellow asparagines), C-terminus motifs (blue), and GPI-anchor attachment signal peptide (red). **B**, Multiple protein alignment of *T. congolense* MISP with MISP360 (MISP-A) and MISP400 (MISP-B). Domains highlighted as in 'A'.
(TIF)

**S4 Fig. Semi-quantitative RT-PCR expression analysis of *misp* mRNA in different trypanosome stages.** BSF, PCF and SG samples shown as in Fig 2A. The $^{\text{HA-GFP}}$MISP360 PCF line either uninduced (360 -) or tet induced (360 +) was introduced expression control. The inset is a representative image of a DNA agarose gel displaying 4 bands corresponding to the RT-PCR amplification of the different *misp* homologs. Expression of the different *misp* isoforms was

normalized to the expression of *tert*. Error bars represent +S.D. from two technical replicates (n = 2).
(TIF)

**S5 Fig. Multiple protein alignment of the 14 BARP homologs in *T. b. brucei*.** Residues colored based on identity conservation from black (maximum) to red (minimum). Alignment made using a Clustal Omega BLOSUM62 matrix with a gap open cost 10 and a gap extension cost of 0.1, using Geneious R9. The detected BARP peptides in the nLC-MS/MS analysis are highlighted in blue. Domains highlighted: ER signal peptides (green), potential N-glycosylation sites (yellow asparagine residues) and GPI-anchor attachment signal peptides (red).
(TIF)

**S6 Fig. Representative immunostaining images.** Non-infected salivary gland (Control); a) *Tbb*-infected (b-e) salivary glands immunostained with either anti-MISP (cyan) or anti-BARP (yellow) polyclonal antibody; DAPI DNA counterstain (magenta); scale bars 200 μm (a, b, c) and 20 μm (d, e).
(TIF)

**S7 Fig. Representative negative controls images.** *T. b. brucei* AnTat 1.1 cultured bloodstream forms (BSF), procyclic cultured form (PCF), midgut procyclic form (PF), proventricular mesocyclic form (MSC), and metacyclic form (MCF) immunostained with anti-MISP polyclonal antibody (cyan); DAPI (magenta); merged and differential interference contrast (DIC); scale bars = 5 mm.
(TIF)

**S8 Fig. Expression of EP-procyclins in proventricular trypanosomes.** Parasites extracted from infected PV at 30 d.p.i. Mesocyclics (MSC), long epimastigote forms (LEMF) and short epimastigote forms (SEMF) probed with anti-EP monoclonal antibody (yellow) and DAPI (blue). Nuclei (N) and kinetoplastids (K) noted in the blue channel. Scale bars: 5 μm.
(TIF)

**S9 Fig. Mean relative fluorescence intensity of anti-body staining.** Anti-MISP (blue) and anti-BARP (orange) staining in attached epimastigote forms (EMF) and metacyclic forms (MCF); bars represent mean values; dots represent individual cell values. error bars indicate ±S.D.
(TIF)

**S10 Fig. Trypanosome populations in infected salivary glands.** Cell type distribution (percentage) in SG-extracted parasites used for the quantification of mean fluorescence intensities of MISP (**A**) and BARP (**B**). Metacyclic forms (MCF), pre-metacyclic forms (P-MCF), epimastigotes with variable kDNA (K) and nuclei (N) numbers (E(1K1N), E(2K1N), E(2K2N)), multi-nucleated epimastigotes (E(multi)) and abnormal cells. **C**, Distribution (percentage) of types of localization of MISP and BARP (**D**), either flagellar (flagellum) or on the whole cell surface (surface).
(TIF)

**S11 Fig. Characterization of salivary gland forms. A**, Schematic of a metacyclic and an epimastigote cell indicating the measurements in 'B': total cell length (red), nucleus to posterior end (orange), nucleus to kDNA (green), and kDNA to posterior end (black); anterior end (A), posterior end (P), kDNA (K), nucleus (N). **B**, Length measurements (mm) described in 'A' in metacyclic (n = 30; gray) and epimastigote cells (n = 30; blue) isolated from tsetse infected salivary glands (from 3 different biological replicates); circles indicate individual data points; thick

horizontal line indicates mean value; error bars indicate ±SD; asterisks indicate significance (*** p<0.001); ns indicates non-significant differences; from one-sided *t*-test assuming normal distribution.
(TIF)

**S12 Fig. Indirect detection of mVSGs in *T. brucei* SG stages.** Immunostaining of epimastigotes (EMF), pre-metacyclics (P-MCF) and metacyclics (MCF) from tsetse infected SG with polyclonal anti-CRD antibody. Nuclei (N) and kinetoplastids (K) noted in the magenta channel. Differential interference contrast (DIC), DAPI (magenta) and anti-CRD (blue). Scale bars: 5 μm.
(TIF)

**S13 Fig. Ectopic localization of tagged MISP360. A**, DNA construct encoding for the ectopic tagged MISP360. **B**, Immunoblotting to detect ectopic (HA-tagged) MISP360 expressed by the mutant HA-GFPMISP360 PCF cell line, either with the transgene uninduced (-) or tet-induced (+), probed with anti-HA (top). PVDF membrane stained with nigrosine after film exposure for sample loading control. **C**, Cellular localization of the HA-tagged ectopic MISP360 in AnTat 1.1 90:13 procyclic (PCF) and metacyclic form (MCF), detected by immunostaining on PFA-fixed non-permeabilized PCF cells with either anti-HA plus anti-MISP (co-localization of ectopic tag with MISP), or with anti-HA plus anti-PFR (flagellar marker) in methanol-fixed cells. Scale bars: 5 μm.
(TIF)

**S14 Fig. Immunodetection of MISP in live SG trypanosomes.** Representative images of live immunostaining of SG parasites using anti-MISP. Epimastigote (EMF) and metacyclic form (MCF) stained with anti-MISP (cyan) and DAPI (magenta). Nuclei (N) and kinetoplasts (K) noted in the blue channel. Scale bars: 5μm. DIC: differential interference contrast.
(TIF)

**S15 Fig. Immunodetection of MISP expression in TSW196 stain trypanosome.** Representative metacyclic cell (MCF) obtained from infected tsetse salivary glands immunostained with polyclonal anti-MISP antibody (cyan); DAPI (magenta); merge, and differential interference contrast (DIC); scale bar = 5 μm. TSW-196 epimastigote and pre-metacyclic cells were also detected by the anti-MISP antibody. Salivary gland stages obtained from three independent tsetse infections with TSW-196 strain were performed and successfully immunostained with anti-MISP polyclonal antibody.
(TIF)

**S16 Fig. Purification of recombinant MISP and BARP proteins. A**, SDS-PAGE (Coomassie blue-stained) of SEC elution fractions of the recombinant MISP360 ectodomain (ladder displaying relative molecular weight on the left (kDa). **B**, Complete SDS-PAGE with fraction of rMISP360 used to determine its crystal structure. **C**, Size exclusion chromatogram of rBARP used for crystallography (peaks for free TRX tag and rBARP monomer indicated). **D**, SDS-PAGE (Coomassie-stained) of rBARP fractions.
(TIF)

**S17 Fig. Comparison of the crystal structure of *T. brucei* MISP360 N-terminus with other trypanosome surface proteins identified as structural analogues.** MISP360 (PDB: 5VTL), BARP (high confidence model), GARP (PDB: Y44), TcHpHbR (PDB: 4E40), TbHpHbR (PDB: 4X0J), VSG 221 monomer (PDB: 1VSG). Structures colored from blue (N-terminus) to red (C-terminus); molecular surface in semi-transparent grey.
(TIF)

**S18 Fig. Suggested models of MISP C-terminal domains. A**, Height comparison between the crystal structure of VSG MiTat 1.2 N-terminus (PDB: 1VSG), the MISP360 N-terminus and the BARP model. **B**, Models of the five *T. brucei* MISP isoforms on the metacyclic surface. mVSGs modelled using the mVAT4 protein sequence and the structures of VSG MiTat 1.2 N- (PDB: 1VSG) and C-terminus (PDB: 1XU6). MISPs modelled using the crystal structure of MISP360 N-terminus.
(TIF)

**S19 Fig. Characterization of *misp* RNAi cells. A**, *In vitro* growth curve of *misp*^RNAi^ PCF cells, either induced (Dox+) or uninduced (Dox-). **B**, Relative *misp* RNA expression in *misp*^RNAi^ PCF cells, midgut procyclics (MG), proventricular parasites (PV) and salivary gland forms (SG). Expression levels of Dox- cells are normalized to 100%. **C**, Representative immunostaining images of non-permeabilized *misp*^RNAi^ MCF (Dox-/+) with anti-MISP (red), DAPI (blue) and phase. D, MISP mean fluorescence intensities (arbitrary units) on *misp*^RNAi^ MCF (Dox-/+). E, Percentage of flies with *misp*^RNAi^ cells infecting the midgut (MG), proventriculus (PV) and salivary glands (SG). Scale bars: 5μm. Error bars show standard deviation, asterisks represent significance (* for p-value < 0.05; *** for p-value < 0.001).
(TIF)

**S20 Fig. Representative images *T. b. brucei* AnTat 1.1 wild type (WT), MISP knockout (KO), and MISP induced addback (AB) trypanosomes.** Cells were immunostained with either anti-CRD polyclonal antibody (A, green), or anti-BARP polyclonal antibody (B, yellow); DAPI (magenta), merged, and differential interference contrast (DIC); scale bars = 5 mm.
(TIF)

**S21 Fig. Schematic of the expression of major GPI-anchored surface glycoproteins throughout the life cycle of *T. brucei*.** Bloodstream forms (BSF), procyclic forms (PF), meso-cyclic forms (MSC), short (SE) and long (LE) epimastigotes infecting the proventriculus, attached epimastigotes infecting the salivary glands (EMF (SG)), pre-metacyclic forms (P-MCF), metacyclic forms (MCF). Representative immunostaining images of the parasite stages highlighting the nucleus and kinetoplast (magenta), and the cell surface (white) (top). Bottom bars define the duration of the expression of the surface proteins in relation to parasite developmental stage.
(TIF)

**S22 Fig. PROSPER protease cleavage site predictions.** MISP360 (top) and MISP400 (bottom) amino acidic sequences with predicted protease cleavage sites highlighted (see key for protease types). Only one member of each MISP subfamily is represented.
(TIF)

**S23 Fig. Characterization of anti-MISP antibody by western blotting. A,** Western blotting of soluble VSG 221 (sVSG), full recombinant MISP380 (rMISP), and recombinant BARP (rBARP) probed with either anti-MISP polyclonal antibody (left), or anti-BARP polyclonal antibody (centre), or anti-VSG 221 polyclonal antibody (right); ladder on the left for apparent molecular weights (kDa). **B,** Blotting membranes developed in 'A' stained with nigrosine. **C,** Instablue-stained SDS-PAGE analysis of same samples as in 'A'.
(TIF)

**S1 File. *T. brucei* BARP, VSG and MISP peptides identified by MS-MS/MS and corresponding supporting spectra.**
(PDF)

**S2 File. Vaccination experiment schedule and western blotting analyses of mouse antisera.**
(PDF)

**S3 File. Quantification of *T. brucei* MISP, BARP, mVSG, and alpha tubulin peptides identified by proteomics analyses in time course RBP6-overexpression experiments performed in Doleželová E. *et al*. (2020) [66].**
(PDF)

**S1 Table. List of *Glossina morsitans morsitans* proteins identified by proteomics analyses in either naïve or *T. brucei*-infected saliva.**
(XLSX)

**S2 Table. List of *Sodalis glossinidius* proteins identified by proteomics analyses in either naïve or *T. brucei*-infected saliva.**
(XLSX)

**S3 Table. List of *Trypanosoma brucei brucei* proteins identified by proteomics analyses in *T. brucei*-infected saliva.**
(XLSX)

**S4 Table. Intracellular *T. brucei brucei* proteins identified by proteomics analyses in *T. brucei*-infected tsetse saliva.**
(DOCX)

**S5 Table. *In silico* predictions of signal peptide, GPI-anchor peptide and *N*-glycosylation sites in all MISP isoforms based on amino acid sequences.**
(DOCX)

**S6 Table. Summary of *T. brucei brucei* VSG species found in either cultured or fly-derived metacyclics according to literature.**
(DOCX)

**S7 Table. Data collection and refinement statistics of TbMISP360 crystal structure.**
(DOCX)

## Acknowledgments

We are grateful to Douglas Lamont and Samantha Kosto (University of Dundee FingerPrints Proteomics Facility) for proteomic analysis and uploading data into PRIDE, and the Biomolecule Analysis and Omics Unit (BAOU) at UTEP/BBRC for performing part of the proteomic data analyses. The authors would like to thank the technical support staff at the Stanford Synchrotron Radiation Lightsource, Drs Federico Rojas and Keith Matthews for generous supply of *T. brucei* AnTat 1.1 Cas9 line, Dr Ian Hastings for statistical advice, and Dr Jay Bangs for kindly supplying aliquots of purified *T. brucei* VSG and anti-VSG antibody. We thank Drs Alena Zíková and Eva Doleželová for helpful discussions and kindly providing MISP expression data in RBP6 cells. We thank members of the Acosta-Serrano group for valuable input and, Dan Southern and Rob Leyland for maintaining the tsetse colony at LSTM.

## Author Contributions

**Conceptualization:** Aitor Casas-Sanchez, Lee R. Haines, Álvaro Acosta-Serrano.

**Data curation:** Aitor Casas-Sanchez, Raghavendran Ramaswamy, Samïrah Perally, Lee R. Haines, Marcela Aguilera-Flores, Cristina Yunta, Igor C. Almeida, Martin J. Boulanger, Álvaro Acosta-Serrano.

**Formal analysis:** Aitor Casas-Sanchez, Raghavendran Ramaswamy, Samïrah Perally, Clair Rose, Laura Smithson, Cristina Yunta, Michael J. Lehane, Igor C. Almeida, Martin J. Boulanger.

**Funding acquisition:** Igor C. Almeida, Martin J. Boulanger, Álvaro Acosta-Serrano.

**Investigation:** Aitor Casas-Sanchez, Raghavendran Ramaswamy, Samïrah Perally, Lee R. Haines, Clair Rose, Susana Portillo, Margot Verbeelen, Shahid Hussain, Laura Smithson, Sue Vaughan, Jan van den Abbeele, Martin J. Boulanger.

**Methodology:** Aitor Casas-Sanchez, Samïrah Perally, Lee R. Haines, Clair Rose, Marcela Aguilera-Flores, Susana Portillo, Margot Verbeelen, Shahid Hussain, Laura Smithson, Sue Vaughan, Jan van den Abbeele, Martin J. Boulanger.

**Project administration:** Álvaro Acosta-Serrano.

**Resources:** Michael J. Lehane, Jan van den Abbeele, Álvaro Acosta-Serrano.

**Software:** Aitor Casas-Sanchez, Igor C. Almeida.

**Supervision:** Michael J. Lehane, Sue Vaughan, Jan van den Abbeele, Igor C. Almeida, Martin J. Boulanger, Álvaro Acosta-Serrano.

**Validation:** Aitor Casas-Sanchez, Raghavendran Ramaswamy, Lee R. Haines, Clair Rose, Marcela Aguilera-Flores, Susana Portillo, Margot Verbeelen, Shahid Hussain, Laura Smithson, Cristina Yunta, Jan van den Abbeele, Igor C. Almeida, Martin J. Boulanger, Álvaro Acosta-Serrano.

**Visualization:** Aitor Casas-Sanchez, Raghavendran Ramaswamy, Lee R. Haines, Clair Rose, Jan van den Abbeele, Martin J. Boulanger.

**Writing – original draft:** Aitor Casas-Sanchez, Álvaro Acosta-Serrano.

**Writing – review & editing:** Aitor Casas-Sanchez, Raghavendran Ramaswamy, Samïrah Perally, Lee R. Haines, Martin J. Boulanger, Álvaro Acosta-Serrano.

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
