## [Decision Letter · Decision Letter 0]

2 Nov 2022

Dear Dr. Acosta-Serrano,

Thank you very much for submitting your manuscript "The Trypanosoma brucei MISP family of invariant proteins are co-expressed with BARP as triple helical bundle structures on the surface of salivary gland forms, but is dispensable for parasite development within the tsetse vector" for consideration at PLOS Pathogens. As with all papers reviewed by the journal, your manuscript was reviewed by members of the editorial board and by several independent reviewers. In light of the reviews (below this email), we would like to invite the resubmission of a significantly-revised version that takes into account the reviewers' comments.

All of the reviewers agree that this paper represents a large amount of high-quality data. The reviewers disagree, however, about whether the results are sufficient for publication in the journal, because you have not actually found any function for these proteins. The decision is based on the premise that - as stated by the more positive reviews - sometimes, negative results are indeed important. In addition, the effects on salivary gland infection might be interesting (although of course, the lack of add-back restoration is a problem that has to be highlighted).

One major issue is clearly the fact that these proteins were previously identified in the Vigneron paper. It is very important that you describe the relevant results from that paper clearly and with sufficient detail in the Introduction, both for the sake of transparency and so that readers can bear in mind the discrepancies as they read your results section.

Since the difference between your results and those of Vigneron et al. could well be caused by the difference between the antibodies used, one more immunofluorescence experiment is essential. This is to use your polyclonal antibody to stain the null mutant, comparing with the wild-type, in order to rule out anti-MISP cross-reactivities (for example, with VSG). (I actually find it odd that you didn't do this already, since you did it for the RNAi.) Only then would the re-naming of these proteins as "MISP" be justified. This, together with at least parts of S16, should be part of the main text; the Figure could also include S8. The extra experiment could also sort out whether the re-expression was insufficient, lines 517-519; and at the same time you can get results concerning the epimastigote/metacyclic balance in the null mutant. (If there is a difference it might be worth checking infectivity to mice, or at least growth in vitro as bloodstream forms.) Assuming the IFA result is as you hope, you should also do the co-staining with anti-BARP and anti-MISP. It is essential to define precisely how you distinguish between procyclics, epimastigotes and metacyclics, and to state how the images were blinded before analysis. Of course if you DO see staining of the null mutant, the whole interpretation would need to be re-assessed.

As noted by one of the reviewers, SDs and significance values with only two replicates are invalid since there is no way to know whether the distribution is normal. Please show the replicates instead and remove the SD and claims of significance. In fact, you should get more data anyway from the essential IFA with the null mutant described above, so you will actually have biological replicates: technical replicates are anyway insufficient. (Note that two technical +two technical doesn't make four replicates!)

all

Obviously all the data needs to be in an appropriate databases - PRIDE or similar for the MS, and PLoS is now asking for all the full gel images with markers (e.g. S11B, S15 C, D, E and F), and all data tables used for Figures, to be in a database such as Figshare as well. I suggest you make sure everything is uploaded before resubmission, on order to minimise later hassle. Once you have done the vital control with the null mutant, please also make sure you have annotated the relevant genes appropriately in TritrypDB.

We cannot make any decision about publication until we have seen the revised manuscript and your response to the reviewers' comments. Your revised manuscript may be sent to reviewers for further evaluation.

Sincerely,

Christine Clayton

Associate Editor

PLOS Pathogens

Margaret Phillips

Section Editor

PLOS Pathogens

Kasturi Haldar

Editor-in-Chief

PLOS Pathogens

orcid.org/0000-0001-5065-158X

Michael Malim

Editor-in-Chief

PLOS Pathogens

orcid.org/0000-0002-7699-2064

Reviewer's Responses to Questions

**Part I - Summary**

Reviewer #1: The manuscript of Casas-Sanchez et al. provides an interesting microbial cell biological and structural characterization of the MISP family of proteins. Significant work goes into identifying the MISP proteins, comparing them, and characterizing them by structural biology. Despite these efforts, the role of the MISP proteins remains as enigmatic at the beginning of the work as by the end of the manuscript. The authors rule out many functions as well as practical applications of the MISP (e.g, use in vaccines) and provide a very solid foundation for further work to understand the role of the proteins in the trypanosome. The experimental work appears very solid (proteomics with MS, crystallography, microbial cell biology, etc). Some of the modeling (e.g, CTD length and conclusions about the MISPs extending beyond the VSGs, possible binding pocket at the top of the molecule, etc), are interesting hypotheses, but not strongly established.

Reviewer #2: The manuscript describes the identification of MISP proteins in infected tsetse saliva and then goes on to characterise their expression, structure and phenotype of a null mutant. Immunization of mice with MISP did not provide protection against infection by tsetse bite.

The mass spec identification of trypanosome proteins in tsetse saliva is fine and probably a technological achievement

The description of the gene family and the description of differences between members is clear.

MISP and BARP mRNA levels were compared by ‘semi-quantitative’ RT-PCR and showed that MISP mRNA was present at highest amounts in the salivary extracts but was also presents in the proventriculus.

This section would be improved by being absolutely clear it is mRNA expression that is being measured and avoiding phrases like ‘Expression of misp genes ..’ (line 256).

A second item that could be clarified is whether the apparent higher abundance of the mRNA encoding the shorter MISPs is due to a shorter amplicon or whether all the amplicons were similar length?

The next experiment looks at MISP and BARP protein expression in different developmental stages using immunofluorescence. No MISP protein was detected in cells from the proventriculus. In the salivary glands, MISP and BARP protein was detected in more than one developmental form with MISP being higher in metacyclics than epimastigotes and BARP the other way round.

As mentioned in the discussion, the expression pattern of MISP is not in agreement with a previous report from another lab (https://doi.org/10.1073/pnas.1914423117). The availability of the MISP null mutant (see below) should be exploited to demonstrate the specificity of the MISP antiserum and resolve the discrepancy.

The structure of one MISP was determined by X-ray crystallography and this showed that the protein adopts a 3-helical bundle structure found in a series of trypanosome surface proteins. This in turn was used to produce a high confidence model of BARP. The work on MISP is fine although the speculation of the pocket being a ligand binding site is just speculation.

The next experiment tested whether pre-immunization with MISP would protect mice against an infected tsetse fly challenge. The experiment is fine and immunization did not protect.

The final set of experiments showed that both an MISP RNAi knock down and a MISP null mutant were able to progress through the salivary glands normally.

The null mutant makes the RNAi experiment redundant.

The manuscript could do with some thorough editing, it comes across as disjointed.

Reviewer #3: While a previous study both mentioned and conducted a few experiments regarding MISP proteins, the experiments conducted were highly different and included a much more rigorous and in depth focus on this particular family of proteins. While the exact timing of expression differs between the two, these can be explained when looking deeper into the differences of the techniques. Aside from the timing of expression inside the salivary gland, investigation into the proteins in the tsetse saliva, the release of surface protein coat into tsetse saliva, the investigation into the knock-out of MISP proteins on fly infections, and the elucidation of the crystal structure of MISP proteins are all novel findings.

The methodology performed in this study was extremely rigorous. The extent of different techniques which were performed showed a truly in-depth and multi-faceted analysis of a surface coat protein family on a mammalian infective stage of Trypanosoma brucei.

Main claims of the paper:

This paper aimed to uncover proteins found in the tsetse fly vector saliva which contribute to transmission of Trypanosoma brucei. The data set which Casas-Sanchez et al. produced was no small feat, as the fly dissections alone for this incredible number of flies (3000!) is quite labor and time intensive. Aside from vector proteins, they found an increase in bacterial symbiont proteins as well as proteins from the trypanosomes themselves, including those found on the surface coat of the parasite. The authors then did a deep dive, implementing a wide array of methods to uncover the mechanism of one of these protein families. They found that the previously poorly described surface proteins, termed Metacyclic Invariant Surface Proteins (MISPs), are found on the surface of epimastigote, pre-metacyclic, and metacyclic cells in increasing amounts as they differentiate in the fly salivary gland. Further, they were able to crystalize and analyze the MISP proteins, showing they have a similar structure not only to BARP on T. brucei but also other uncharacterized coat proteins in related species. Less visually displayed in their paper but no less important was the deletion and add back of all of the MISP proteins from the parasite to check its role in infecting tsetse flies as well as probing these surface proteins as a potential vaccine target (though using a MISP protein as a vaccine candidates had been done previously in Aksoy et al.the end results were confirmed). Though both experiments resulted in negative data, the experiments are no less important and are still telling in that we now know the protein family is not essential to protein colonization of the salivary gland and are not suitable vaccine targets, at least using GLA-AF.

It is important to note that a previous study, Vigneron et al. PNAS 2020, also looked at one MISP protein (termed SGE-1), albeit with different methods, which they had found during an RNA-seq experiment. Vigeron et al. found that the protein is expressed during a different life cycle stage. They showed that a MISP (or SGE-1) protein is expressed only by epimastigotes in the salivary glands while Casa-Sanchez et al. show that MISP expression begins in epimastigotes and then expression increases as they become metacyclics, while BARP expression is the exact opposite.

Casa-Sanchez et al. offer 3 reasons for this divergence in results.

The first reason includes the fundamental differences in any experimental set up. Namely, in strains and conditions used. Vigeron et al. used the RUMP 53 strain and looked at salivary gland infection from 30-33 dpi using only female flies. Casas-Sanchez et al. used the TSW196 strain and looked at salivary gland infection 28-30 dpi using only male flies. As Casa-Sanchez et al. showed that these conserved MISP proteins are non-essential, this may have allowed expression differ between the strains. This, however, would be quite easy to test, especially as Casa-Sanchez et al. use chemicals to increase infection rate, lowering the number of flies they would need to use. We would recommend that they infect flies with a different strain of trypanosome and perform immunofluorescence to check that the expression pattern of MISP matches what they saw with TSW196, rather than only the epimastigote stage as Aksoy et al. claim.

Difference in infection days was another reason offered for the difference and this could also be directly tied to the fly sexes used. The trypanosome field normally uses male flies when looking at infection processes, as female flies are known to be less easily infected with T. brucei. As the difference in infection rate between the sexes is already known (though the mechanisms of which are hypothesized but not fully understood), this could also cause differences in trypanosome development timing. While possible, we find that this reason is unlikely as the different stages of trypanosomes should still have a similar expression pattern whether the fly has been infected for 28 days or 33 days. However, both papers are somewhat unclear about how they identified epimastigotes vs. pre-metacyclic vs. metacyclics. Both Casa-Sanchez et al. and Vigeron et al. do not clarify their method of differentiating between the stages. While it seems obvious that this was based on morphology, kinetoplast – nucleus placement, and attachment, adding a sentence about exactly how the authors defined the stages would help clear this up. This would especially be helpful as Vigeron et al. does not have a pre-metacyclic stage included in their protein results and could be another source of the difference of results.

In terms of fly timing, as Casa-Sanchez et al. have already laid out solid experiments with clear data and hypothesis’, looking further into differences in the individual fly infection days in which the non-essential MISP proteins have different expression would just not be feasible as the experiments would take too long and be too labor intensive when there is a high potential that this was not the cause of the difference.

In my view, the next two explanations given for the differences between the two papers results hold a lot more merit. The reasons being that transcription does not equal translation and that monoclonal antibodies can experience problems binding epitopes which can be blocked by tightly packed proteins, such as the VSG coat. While RNA-sequencing is a very powerful tool to look at gene expression, final protein cannot be confirmed until protein studies are done. Vigeron et al. did follow up looking at the MISP proteins by using immunofluorescence but used monoclonal antibodies which were made against whole protein, with the unknown epitope being anywhere on the protein. As Vigeron et al. only used non-permeabilized cells for the antibody work, it is very possible that the mVSG on the metacyclics blocked binding in metcyclics and not in the epimastigotes which have just started to express MISP. Meanwhile, Casas-Sanchez et al. used polyclonal antibodies, used both permeabilized and non-permeabilized cells, as well as live cells to confirm expression.

Reviewer #4: The authors did an analysis of tsetse saliva to identify secreted proteins. They identify a single peptide of Tb927.7.360, which belongs to a gene family of 5 proteins. Despite being 1 peptide, the single MS/MS spectra for this peptide looks very convincing. They follow the expression of these genes in different stages by a modified RT-PCR protocol and IF. They obtained a crystal structure for parts of TbMISP-360. Functionally, the proteins do not protect against T.brucei infection and are not essential for colonization of the SG.

**Part II – Major Issues: Key Experiments Required for Acceptance**

Reviewer #1: (No Response)

Reviewer #2: One experiment only: use of the MISP null mutant to demonstrate specificity of MISP antiserum.

Reviewer #3: none

Reviewer #4: The manuscript is well-written and nicely structured. Although a lot of work by the authors, it unfortunately remains descriptive due to a lack of functional understanding. I would consider it a strong manuscript for publication here, if there would be any insight into its function. At the moment, I feel the manuscript is more suited for a more specialized journal.

To gain insight, omics methods might be of help. The authors tested very specific scenarios of their protein functionality. How about performing transcriptomics/proteomics in the knockout strains to shed light on changes during SG infection or after establishment in the vertebrate skin. Following more targeted, the authors suggest a functionality in skin colonization. This could actually be checked experimentally.

For the crystal structure, maybe the authors could integrate folding predictions from alphafold to gain also ideas about the other members of the protein family and to further support their homology fold model of the misp360. I also wonder whether the authors could further substantiate their claim that especially MISP-A is extending beyond the VSG coat (Figure 3H).

The functional characterizations are unfortunately not leading to a further characterization or even a lead on the function of the proteins. However, I wonder why the authors just reintroduced MISP360 and not a cassette with all misp genes as this might have led to a rescue of the albeit small, but significant salivary gland infection rates.

The protein is obviously expressed in metacyclics generated by overexpression of RBP6. This could be utilized to study the metacyclics in the in vitro culture system. Do they show differences during the differentiations? These systems are readily accessible for growth, differentiation, transcriptomics and proteomics…

The authors do a very extensive discussion, but some experiments are rather easy to perform and could contribute to a more solid characterization of the new protein. E.g. the mentioned skin infection behavior or the inclusion of the proteins in EVs.

**Part III – Minor Issues: Editorial and Data Presentation Modifications**

Reviewer #1: (No Response)

Reviewer #2: There has been a previous study on MISPs that gave them a different name and came to different conclusions about the expression pattern. This is addressed above but I feel the authors should make this clearer, probably in the introduction as well as the discussion.

Reviewer #3: Adding another sentence into the methods stating how many flies were used per replicate when collecting the fly saliva of infected vs non-infected tsetse.

For Figure 1 C. the misp-B arrows are purple rather than dark blue, as stated in the figure legend (line 202-203)

Line 260-261, ‘in parallel’ should be moved to the beginning of the sentence for clarity.

Add immunofluorescence images of BARP and MISP co-expressing. Line 368 and S17Fig clearly states that BARP and MISP are co-expressed, but the IF images shown (Figure 2) either show one or the other on a given cell at a given time, not true co-expression.

Add the ladder to the western blots images

Line 495, change ‘developed’ to ‘developing’

Line 496, change ‘either’ to ‘both’ and add an ‘s’ to ‘antigen’

Line 504, change ‘neither’ to ‘either’ and ‘nor’ to ‘or’

While hidden in the supplements, we think the ‘small’ but significant reduction of salivary gland infection in the MISP knock out cell lines is very compelling. It would be interesting to look at different numbers of epimastigotes vs. metacyclics in the salivary glands using these knock out lines (as S16 Fig H only looks at general salivary gland infection – not type or numbers) or to see if the knock out metacyclics have any other differences in vitro (rather than just parasitemia in the mouse) but these would again take a lot of time and are not essential to this paper. Instead, they would be nice follow up experiments done at a later date.

Reviewer #4: 1) Figure 1C: the purple for the repeats and for the MISP-B is not intuitive, maybe change the color of either one to make a differentiation.

2) Figure 2B, 2C, 2E: the image space could be used better, some of the trypanosomes are pretty small while most of the area is black (done better in 2D)

3) Figure 2B, 2C, 2E: a negative control with PCF, BSF or MG stage is necessary for showing antibody specificity (similar to Fig S6).

4) Figure 2A: significance from two technical replicates is not acceptable, three if possible biological replicates should be used. Why is it a one-sided t-test? Did the authors expect to just see upregulation to begin with?

5) Figure 3b: the inlet plot should be covering the size range of 10 – 200 kD to judge the purity of the fraction, otherwise this is not very informative.

6) Figure S4: Why is the normalization done to tert and not a universal expressed transcript?

7) Material and Methods: line 757 there is an alkylation step missing, otherwise carboxyamidomethylation as a fixed modification in MS/MS search does not make sense

8) MM: line 764, did you really search for acetylation on ANY amino acid as a variable modification? This does not make any sense chemically and is also not a preset standard in the field. I guess this just got wrongly described here, otherwise I would be curious for an explanation of this setting.

9) MM line 769ff, for all downloaded databases please indicate the triTrypDB database version instead of the downloaded date. They do versioning and this is easier to reproduce.

10) The original mass spec data should be uploaded to a repository like PRIDE.

PLOS authors have the option to publish the peer review history of their article (what does this mean?). If published, this will include your full peer review and any attached files.

Reviewer #1: No

Reviewer #2: No

Reviewer #3: No

Reviewer #4: No
---

## [Decision Letter · Decision Letter 1]

8 Mar 2023

Dear Dr. Acosta-Serrano,

We are pleased to inform you that your manuscript 'The Trypanosoma brucei MISP family of invariant proteins is co-expressed with BARP as triple helical bundle structures on the surface of salivary gland forms, but is dispensable for parasite development within the tsetse vector' has been provisionally accepted for publication in PLOS Pathogens.

Best regards,

Christine Clayton

Academic Editor

PLOS Pathogens

Margaret Phillips

Section Editor

PLOS Pathogens

Kasturi Haldar

Editor-in-Chief

PLOS Pathogens

orcid.org/0000-0001-5065-158X

Michael Malim

Editor-in-Chief

PLOS Pathogens

orcid.org/0000-0002-7699-2064

All concerns of the reviewers have now been addressed.

Reviewer Comments (if any, and for reference):

Reviewer's Responses to Questions

**Part I - Summary**

Reviewer #2: The manuscript describes the discovery and characterisation of MISP proteins expressed on the cell surface of metacyclic stage trypanosomes. It goes on to show that they are not successful when trailed as transmission blocking vaccines.

I reviewed the manuscript previously and am happy that the new version has answered all my concerns and is now an interesting, thorough and informative piece of work.

Reviewer #3: The authors have addressed all of my concerns. They have added an experiment showing MISP expression in another strain (TSW196, Fig S15), added a more clear explanation of their identification methods for the different SG forms (Fig S11), and rectified other small details I believed necessary to add.

I would like to re-state that the work done in this paper is clearly rigorous and gives insight into the trypanosome infective stage, even if the main protein studied was non-essential to trypanosome infection of the tsetse. The results and following discussion give deeper insight into the factors present in infected tsetse saliva as well as a detailed description of a previously less known surface protein for the infective metacyclic stage of trypanosomes (MISP).

This paper is ready for publication.

Reviewer #4: The authors satisfactory addressed all my comments.

**Part II – Major Issues: Key Experiments Required for Acceptance**

Reviewer #2: (No Response)

Reviewer #3: (No Response)

Reviewer #4: None.

**Part III – Minor Issues: Editorial and Data Presentation Modifications**

Reviewer #2: (No Response)

Reviewer #3: (No Response)

Reviewer #4: None.

PLOS authors have the option to publish the peer review history of their article (what does this mean?). If published, this will include your full peer review and any attached files.

Reviewer #2: No

Reviewer #3: **Yes: **Markus Engstler

Reviewer #4: No

---

## [Editor Report · Acceptance letter]

27 Mar 2023

Dear Dr. Acosta-Serrano,

We are delighted to inform you that your manuscript, "The Trypanosoma brucei MISP family of invariant proteins is co-expressed with BARP as triple helical bundle structures on the surface of salivary gland forms, but is dispensable for parasite development within the tsetse vector," has been formally accepted for publication in PLOS Pathogens.

Best regards,

Kasturi Haldar

Editor-in-Chief

PLOS Pathogens

orcid.org/0000-0001-5065-158X

Michael Malim

Editor-in-Chief

PLOS Pathogens

orcid.org/0000-0002-7699-2064